

# Lithology and climate controlled soil aggregate size distribution and organic carbon stability in the Peruvian Andes

Songyu Yang [1], Boris Jansen [1], Samira Absalah [1], Rutger van Hall [1], Karsten Kalbitz [2], Erik Cammeraat [1]

5    1 Institute for Biodiversity and Ecosystem Dynamics, University of Amsterdam, Amsterdam, Netherlands

2 Soil Resources and Land Use, Institute of Soil Science and Site Ecology, Technische Universität Dresden, Dresden, Germany

Correspondence to: Songyu Yang (s.yang@uva.nl; longxianfeijian@163.com)



**Abstract**

Recent studies indicate that climate change influences soil mineralogy by altering weathering processes, and thus impacts soil aggregation and organic carbon (SOC) stability. Alpine ecosystems of the Neotropical Andes are characterized by high SOC stocks, which are important to sustain ecosystem services. However, climate change in the form of altered precipitation patterns can potentially affect soil aggregation and SOC stability with potentially significant effects on the soil's ecosystem services. This study aimed to investigate the effects of precipitation and lithology on soil aggregation and SOC stability in the Peruvian Andean grasslands, and assessed whether occlusion of organic matter (OM) in aggregates controls SOC stability. For this, samples were collected from limestone soils (LSs) and acid igneous rock soils (ASs) from two sites with contrasting precipitation levels. We used a dry-sieving method to quantify aggregate size distribution, and applied a 76-day soil incubation with intact and crushed aggregates to investigate SOC stability in dependence on aggregation. SOC stocks ranged from $153\pm27$ to $405\pm42$ Mg ha$^{-1}$, and the highest stocks were found in the LSs of the wet site. We found lithology rather than precipitation to be the key factor regulating soil aggregate size distribution, as indicated by coarse aggregates in the LSs and fine aggregates in the ASs. SOC stability estimated by specific SOC mineralization rates decreased with precipitation in the LSs, but minor differences were found between wet and dry sites in the ASs. Aggregate destruction had a limited effect on SOC mineralization, which indicates that occlusion of OM in aggregates played a minor role in OM stabilization. This was further supported by the inconsistent patterns of aggregate size distribution compared to the patterns of SOC stability. We propose that OM adsorption on mineral surfaces is the major OM stabilization mechanism controlling SOC stocks and stability. The results highlight the interactions between precipitation and lithology on SOC stability, which are likely controlled by soil mineralogy in relation to OM input.

**Keywords:** soil organic matter; stabilization; precipitation; limestone; acid igneous rock; aggregate destruction

## 1 Introduction

Soil organic carbon (SOC) is the largest terrestrial carbon (C) pool and plays an important role in global C dynamics (Carvalhais et al., 2014; Lal, 2004). However, the distribution of SOC at a global scale is highly variable (Batjes, 2014; Lal, 2004). Alpine grasslands of the Andes are characterized by large SOC stocks, and play a crucial role in agricultural production, water provision and sustaining high biodiversity





(Buytaert et al., 2011; Muñoz García and Faz Cano, 2012; Rolando et al., 2017a; Tonneijck et al., 2010). The large SOC stocks contribute to crucial ecosystem services, and act as a potential C sink or source for atmospheric $CO_2$ in the context of global change (Buytaert et al., 2011). However, the Andean region is characterized by heterogeneity in climate, vegetation, agriculture activities and geological formations (Buytaert et al., 2006b; Rolando et al., 2017a), which potentially introduces spatial variability in SOC storage and stability.

Recent views on SOC persistency have shifted from chemical recalcitrance of soil organic matter (OM) to progressive decomposition of soil OM dependent on the surrounding biotic and abiotic environment (Lehmann and Kleber, 2015; Schmidt et al., 2011). Specifically, SOC storage is controlled by soil OM stabilization, which in the emerging paradigm is dominated by (1) OM adsorption on mineral surfaces that controls long-term stabilization, and (2) physical occlusion of OM within soil aggregates that regulates intermediate-term stabilization with heterogeneous OM composition and residential time (Lützow et al., 2006; Schrumpf et al., 2013). Adsorption of OM on mineral surfaces was reported as an important stabilization mechanism for soil OM underlying the large SOC stocks in the Andes (Buytaert *et al.*, 2006a; Tonneijck *et al.*, 2010; Yang *et al.*, submitted). However, studies focusing on aggregate-controlled OM stabilization in relation to climate in the Andes are rare (e.g. Rolando *et al.*, 2017b). Aggregates promote soil OM stabilization against decomposition by regulating the availability of oxygen and water as well as the accessibility of OM itself (Kong et al., 2005). Thus, the formation and turnover of soil aggregates are crucial for SOC storage and OM stabilization (Six et al., 2004; Six and Paustian, 2014). As soil aggregates are formed with monomers of clay minerals, polyvalent cations and OM, their formation and the underlying OM stabilization largely depend on various biotic and abiotic factors (e.g. climate and lithology) (Bronick and Lal, 2005; Doetterl et al., 2015).

Lithology is the key factor controlling SOC storage and OM stabilization, mainly attributed to its controls on soil mineralogy and texture (Angst et al., 2018; Homann et al., 2007; Wiesmeier et al., 2019). In soils formed on acidic bedrocks, OM is generally considered to be stabilized by ligand exchange with non-crystalline Fe and Al oxides, whereas in soils formed on alkaline-rich bedrocks, OM is thought to be stabilized by interaction with the mineral surface through polyvalent cation bridges (e.g. $Ca^{2+}$) (Lützow et al., 2006). Soil texture also has effects on OM stabilization because OM-mineral association is dominantly controlled by clay-sized minerals (Kaiser and Guggenberger, 2003; Kleber et al., 2007). In addition, soil mineralogy and texture are crucial factors for aggregate formation, which potentially regulates soil OM stabilization as controlled by occlusion of OM within aggregates (Bronick and Lal, 2005; Six et al., 2004).



Climate factors, comprising temperature and precipitation, act as the primary drivers regulating SOC

storage and OM stabilization by controlling OM input and decomposition (Schmidt et al., 2011; Wiesmeier et al., 2019). Recent studies indicated that climate factors also control OM persistence by regulating soil mineralogy (Chaplot et al., 2010; Doetterl et al., 2015, 2018). The soil mineralogy and OM persistence controlled by climate can be dependent on lithology due to their inherent properties (Jenny, 1994; Wagai et al., 2008). Given the importance of climate and lithology, the heterogeneity in

precipitation and lithology in the Andes can potentially cause spatial variation in OM stabilization and consequently SOC stocks. In addition, shifts in e.g. precipitation patterns as a result of global change may impact SOC stocks in different parts of the Andes in different ways.

The objectives of our study were to assess the controls of precipitation and lithology on SOC stocks and stability in the Peruvian Andes. Specifically, we aimed to investigate whether the effects of precipitation

and lithology on SOC stability are through the controls of OM stabilization governed by aggregate occlusion and/or mineral adsorption. For this, we applied a combination of aggregate-size fractionation with a 76-day incubation for soil samples collected from the Peruvian Andes with two contrasting bedrocks and two precipitation levels.

**2 Materials and methods**

*2.1 Site description*

Basic information of the sampling sites is shown in Fig. 1. The study areas belong to the Neotropical alpine grassland of the Andes, corresponding to the ecosystem of wet Puna or Jalca (Rolando et al., 2017a). Two sampling sites were selected with similar altitudes but with different lithologies and

precipitation levels. The wet site is located in the western Cordillera mountain chain of the Peruvian Andes, to the west of Cajamarca, Peru (7°11` S, 78°35` W). The altitudes of the sites range from 3500 m to 3720 m asl. The temperature shows a large daily variation and minor seasonal variation, with an estimated annual mean of 11°C. The sites receive 1100mm precipitation per year and have a wet season between October and April (Sánchez Vega et al., 2005). The dry site is located in the mountain chain of

the Cordillera Blanca, to the northeast of Carhuaz (9°22` S, 77°59` W), with altitudes ranging between 3490 and 3700 m asl. The annual temperature and precipitation were estimated as 11 °C and 680 mm, and had similar annual and daily variations as the wet site (Merkel, 2017). Typical land use in both sites is grassland with human activities including cultivation, grazing and plantation of pine trees and eucalyptus (Rolando et al., 2017a; Sánchez Vega et al., 2005).





For the wet site, the geology consists of a basement of Cretaceous sedimentary formations, which is composed of limestone, marl, shale and quartzite. Neogene igneous bedrocks consisting of granite and ignimbrite intrude or cover parts of the basement (Reyes-Rivera, 1980). For the dry site, intrusive igneous rocks (mainly granodiorite) belonging to the Neogene Cordillera Blanca batholith are present in the western part of the Cordillera Blanca (Coldwell et al., 2011; Portes et al., 2016). The foot slopes consist

of fluvio-glacial and glacial sediments partly covering andesitic ignimbritic rocks of the Neogene Yungay Formation, as well as the sedimentary Cretaceous Carhuaz and Santa formations that are dominated by folded limestones, sandstones and shales (Coldwell et al., 2011). Soils developed on the limestone were classified as Phaeozems or Umbrisols, whereas soils on acid igneous rocks were classified as Andosols and Umbrisols (WRB, 2014).


### 2.2 Sampling procedures

For both the wet and dry sites, we selected three soil sampling plots from limestone and three plots on acid igneous rocks. For limestone soils (LSs) in both sites and acid igneous rock soils (ASs) in the wet site, all soils were directly developed on the bedrock. For ASs in the dry site, one sampling site was

directly developed on granodiorite, whereas the other two sites were located on the glacier deposits on lateral moraines with a granodioritic composition. All sampling sites were selected based on the criteria of (1) grassland, grassland with shrubs or abandoned cropland, (2) gentle slopes, (3) no intensive human activities, and (4) similar soil development status.

For the determination of bulk density and calculation of SOC stocks through the soil profile, samples

were collected every 10 cm in duplicate to the depth of the C horizon using Kopecky rings (100 $cm^3$). For the determination of basic soil properties, aggregate-size fractionation and incubation, soil samples were collected per horizon in triplicate (e.g. $A_{h1}$, $A_{h2}$ and $B_{tg}$ horizons). To minimize aggregate destruction during transportation, soil samples were transferred into sealed plastic bags and protected by hard plastic boxes.


### 2.3 Laboratory analyses

Soil samples for the determinations of bulk density and SOC stocks were freeze-dried after removing gravels (>2 mm). Soil moisture contents and bulk densities were measured by weighing samples before and after freeze-drying. Soil samples collected per horizon were air-dried, followed by taking 5-10 g of



sub-samples milled for the determination of basic soil properties. For these samples, total C and N

contents were analyzed using a VarioEL Elementar analyzer (Elementar, Germany). As inorganic C

contents were negligible in all the samples, the total OC contents were equal to total C contents. Soil pH

was determined with a glass electrode in suspensions of soil material in demi-water (w:v=1:5, Bates,

1973).

Total SOC stocks were calculated using the following equation:

$$SOC\ stock = \sum_{i=1}^{i=k} BD_i \times C_i \times (1 - S_i) \times D_i$$

In which, $BD_i$ = bulk density (g cm$^{-3}$) of the layer i, $C_i$ = SOC content (%) of the layer I, $S_i$ = stoniness (%)

of layer I, $D_i$= thickness (cm) of layer i.

Dry-sieving was applied to fractionate soil samples into 5 aggregate-size groups: >5mm, 2-5mm, 0.22-2

mm, 0.063-0.25 mm and <0.063 mm, respectively. Briefly, 170-230 g sub-samples (<16 mm) of each

horizon were fractionated using 4 mesh sieves (5, 2, 0.25 and 0.063 mm) by shaking for 20 s at 30 Hz at a

horizontal shaker. Gravel contents (>2 mm) were calculated for all fractions larger than 2 mm and

removed. For each fraction, fraction weights as well as total C and N contents were determined. The mean

weight diameter (MWD) of the bulk soil was calculated by:

$$MWD = \sum_{i=1}^{i=5} \frac{x_{i\ max} + x_{i\ min}}{2} \times w_i$$

In which, $x_{i\ max}$ = maximum diameter (mm) of the fraction i, $x_{i\ min}$ = minimum diameter (mm) of the

fraction i, $w_i$ = weight percent of the fraction i (Klute and Dinauer, 1986).

Sample materials collected from different horizons were used for the incubation. All materials from

individual A horizons in the same soil profile were merged (e.g. Ah1 and Ah2 horizons merged to A

horizon), based on the weight distribution of the horizons as estimated by their bulk densities and depths.

Original B horizons were used because each soil profile only had a single B horizon. Prior to the

incubation, all samples were fractionated into large macroaggregates (LM, > 2mm), small

macroaggregates (SM, 0.25-2 mm) and microaggregates (Mi, <0.25 mm), following the dry-sieving

procedure. The LM and SM fractions were used for the incubation with intact and crushed aggregates.

The variation in SOC mineralization between intact and crushed aggregates was used as a measure of C

stabilization by occlusion within aggregates (Goebel et al., 2009). Aggregates were crushed by grinding

the fractions using a porcelain mortar, and all crushed materials could pass a 0.125 mm sieve (Wang et al.,

2014). Before incubation, intact and crushed fractions were rewetted at pF 2.0 for 10 days to activate soil





microbes. Approximate 10 g dry-weight equivalent fractions were incubated for 76 days at 20 °C in

sealed glass jars (120 ml). All soil fractions were incubated in duplicate. The headspace of incubating jars was sampled on days 1, 2, 6, 9, 13, 20, 28, 48 and 76. During the sampling period, $CO_2$-free air was injected into the jars to maintain pressure and avoid too high $CO_2$ concentrations. The $CO_2$ concentration was analyzed using a gas chromatograph with a flame ionization detector (GC-FID, Thermo Scientific, Trace GC Ultra) with packed columns (RESTEK Packed Column, Part Nbr: PC7130, Serial Nbr:

C34216-01, HayeSep Q, 1/8" 80/100 2m and HayeSep Q, 1/8" 80/100 1m). A methanizer was situated in front of the FID, as the detector can only measure hydrocarbons instead of $CO_2$. Specific SOC mineralization rates (g $CO_2$-C $g^{-1}$ C) were used as an indicator of the C stability of the soil fractions.

### 2.4 Statistics

Statistical comparisons of soil properties and SOC stocks were made using a one-way ANOVA. *Post hoc* analyses were conducted using the Fisher's Least Significant Difference (LSD) test. Principal component analysis (PCA) was applied to investigate potential differences between different soil profiles and horizons. Before conducting the PCA, Kaiser-Meyer-Olkin tests and Bartlett's tests were used to guarantee that sampling adequacy and the sphericity were suitable for the analysis. Linear regressions

were applied to investigate relationships of specific SOC mineralization rates with SOC and C/N ratios. An independent T-test was applied to check effects of precipitation, lithology, soil horizon, aggregate size and aggregates destruction on SOC mineralization rates.

Before the T-test and the one-way ANOVA, data normality and variance homogeneity were examined using a Shapiro-Wilk test and a Levene's test. When the assumption of normality was violated, the

Kruskal-Wallis H test was applied instead of the one-way ANOVA, while the Mann-Whitney U-test was used instead of the T-test. When the homogeneity of the variance could not be assumed, the Robust Welch test was used for the one-way ANOVA. All analyses were conducted using SPSS 24.0 (SPSS Inc., USA).

**3 Results**

### 3.1 Soil properties

Average soil depths were 61cm for limestone soils (LSs) in both wet and dry sites, and 49 cm and 51 cm for acid igneous rock soils (ASs) in the wet and the dry sites (Fig. 1). SOC stocks were highest in LSs of





the wet site (wet-LSs, 405.3 ± 41.7 Mg ha$^{-1}$), followed by ASs of the wet site (wet-ASs), ASs of the dry
site (dry-ASs) and dry-LSs. SOC stocks in the wet-LSs were significantly higher compared to other soils
(Fig. 2). SOC contents in the A horizons were significantly higher in the wet-LSs both with regard to
bedrock and precipitation. No significant differences were present for the ASs with regard to precipitation
(Fig. 2). The LSs had no significant difference in C/N ratios compared to the ASs for the A horizons in
the wet sites, however, the LSs had significantly lower C/N ratios in the dry site (Fig. 2). With decreasing
precipitation, C/N ratios significantly decreased in the LSs and increased in the ASs (Fig. 2). pH values
were significantly higher in the LSs compared to the ASs in the wet site, but were not significantly
different in the dry site (Fig. 2). In addition, significantly lower pH values with lower precipitation were
only found in the LSs (Fig. 2). With regard to the differences between horizons in the LSs, B horizons
were characterized by significant lower SOC contents, lower C/N ratios and higher pH compared to A
horizons, except for SOC contents and pH values in the dry sites (Fig. 2).

### 3.2 Aggregate-size fractionation

In both wet and dry sites, LSs having larger aggregate sizes than ASs (Fig. 3A and 3B). The LSs had
larger contributions of fraction materials and OC from the LM fraction (>60 %) and minor contributions
from the Mi fraction (<10 %, Fig. 3C and 3D). In contrast, the ASs had larger contents of the SM and the
Mi fractions (Fig. 3C and 3D). When comparing the wet and dry sites, LSs were not different, whereas
wet-ASs were slightly different from the dry-AS, as shown by larger aggregate sizes and more OC
present in large aggregates (Fig. 3A-3D). When comparing the A and B horizons in the LSs, B horizons
had larger aggregates fractions weights and more OC in the LM compared to the A horizons (Fig. 3C and
3D).

Soil properties of different horizons are shown in Fig. 4. PC1 and PC2 explained 67.0 % and 17.9 % of
the total variation. PC1 had positive contributions of the SM and Mi fractions and negative loadings of
the LM fractions and MWD, whereas PC2 had positive contributions of C and N contents. The LSs were
separated from the ASs as indicated by coarser aggregates, higher pH values and lower C/N ratios (Fig. 4).
In addition, wet-LSs were separated from dry-LSs by higher C and N contents, whereas ASs were not
clearly separated by precipitation (Fig. 4). The LSs were characterized by increasing coarse aggregate
fractions and decreasing C and N contents as well as C/N ratios with increasing soil depth, whereas the
ASs had no clear pattern in soil property change with increasing depth (Fig. 4).





*3.3 SOC mineralization*

After the 76-day incubation, specific SOC mineralization rates were the highest in A and B horizons of the dry-LSs, when compared to the other soil horizons (Fig. 5A-5D). For comparisons between two lithologies, SOC mineralization rates were not significantly different in the wet site, but were generally higher in the LSs compared to the ASs in A horizons of the dry site (Table 1). For effects of precipitation,

SOC mineralization rates were significantly higher in the dry site compared to the wet site for the LS-A horizons in most sampling days, but were not significantly different for the AS-A horizons (Table 1). For comparisons between A and B horizons in the LSs, SOC mineralization rates were not significantly different in the wet site. In the dry site, A horizons had significantly higher SOC mineralization rates than B horizons only in the aggregate-crushed SM fraction (Table 1).

SOC mineralization rates were slightly stimulated (up to 19.4 %) when aggregates were crushed compared to that when aggregates were intact, with exceptions of the LM fraction in dry-AS-A horizons and the SM fraction in dry-AS-A horizons and wet-LS-A horizons (Fig 6A and 6B). However, the stimulation caused by aggregate destruction was never significant (Fig. 6A and 6B). In addition, no significant difference in SOC mineralization rates was found between LM and SM fractions. Exclusively,

slightly higher SOC mineralization rates (not significant) were found in the SM fraction compared to LM fraction in A horizons of the wet-LS, the dry-LSs and the dry-ASs (Fig. 6C and 6D).

Overall, SOC mineralization rates had significant negative relationships with SOC contents and C/N ratios, and the negative relationships did not differ between intact and crushed aggregates (Fig. 7A and 7B). Exclusively for the dry-LSs, positive relationships were found between SOC mineralization rates and

SOC contents when aggregates were intact and crushed, and between SOC mineralization rates and C/N ratios when aggregates were crushed (Fig. 7C-7F). In the dry-LSs, SOC contents and C/N ratios explained 38.2 % and 24.9 % of the variation of specific SOC mineralization rates when aggregates were intact. When aggregates were crushed, SOC contents and C/N ratios explained 48.0 % and 33.3 % of the total variation (Fig. 7C-7F).


**4 Discussion**

*4.1 Aggregate size distribution*

Lithology is the key factor controlling soil aggregate size distribution in our soil samples. This is indicated by a coarser aggregation and more SOC present in large-sized aggregates for the LSs, and a



finer aggregation and more SOC present in fine aggregates for the ASs (Fig. 3 and 4). Lithology generally

controls soil aggregation by affecting soil mineralogy (Bronick and Lal, 2005). As the LSs are developed

from calcareous bedrocks, their high calcium (Ca) and clay contents play an important role in promoting

the formation of large-sized aggregates (Bronick and Lal, 2005; Six et al., 2004). In contrast, the ASs

developed on acid bedrocks with a lower pH (Fig. 2) and less alkaline cations (e.g. $Ca^{2+}$) present. As

alkaline cations like $Ca^{2+}$ act as polyvalent cation bridges to promote soil aggregation (Boix-Fayos et al.,

2001; Bronick and Lal, 2005), the lack of alkaline cations hinders the formation of coarse aggregates in

the ASs. In addition, the coarse soil texture of the ASs (Yang *et al.*, submitted) also has negative impacts

on the formation of larger aggregates. The controls of lithology can be further supported by

physicochemical properties of each aggregate fraction, which showed that the ASs were distinguished

from the LSs by (1) having the lowest fraction C contents and ratios of fraction to bulk soil C contents in

the SM fraction, and (2) decreased C/N ratios with aggregate size in the fractions < 2 mm (Fig. S1).

Furthermore, increasing aggregate sizes with soil depth were found in the LSs exclusively (Fig. 4), which

can be explained by the better aggregation promoted by clay illuviation in deep soils. In contrast, no clear

vertical differences in the ASs may be related to the lack of the clay fraction in the ASs (Yang *et al.*,

submitted).

Unlike lithology, precipitation plays only a minor role in the soil aggregate size distribution for our soils.

This is indicated by small differences in properties related to soil aggregation between the wet and the dry

sites for the same bedrock types (Fig. 3, Fig. 4 and Fig. S1). Potential effects of precipitation on soil

aggregate formation could be present because of the variation in vegetation and OM input (Bronick and

Lal, 2005). Both wet and dry sites are parts of the alpine grassland zone (Puna), which had comparable

vegetation and OM input source. Although precipitation can potentially control the quantity and quality of

the OM input (Wiesmeier et al., 2019), the extent of the OM input variation may be insufficient to alter

the soil aggregation in our study. Notably, the dry-ASs had finer aggregates than the wet-ASs (Fig. 3).

This is probably attributed to their greater gravel contents in the LM fraction (Fig. 1 and Table S1), which

is likely related to the steep mountains and the glacier materials in the dry-AS sampling site rather than

precipitation (Portes *et al.* 2016). Nevertheless, similar properties of aggregate fractions between wet-ASs

and dry-ASs (Fig. 4 and Fig. S1) still suggest that precipitation has minor controls in aggregation patterns

of the AS.

*4.2 SOC stocks and stability*





SOC stocks were controlled by interactions between lithology and precipitation, as indicated by increased stocks with precipitation in the LSs and no significant changes in the ASs (Fig. 2). Lithology had significant effects on SOC stocks in the wet sites (Fig. 2), which is consistent with the findings of Yang *et al.* (2018) showing that lithology is the key factor controlling SOC stocks. In the wet site, the high SOC

stocks in the LSs compared to the ASs can be explained by deeper soils and higher SOC contents in A horizons (Fig. 1 and 2). In the dry site, no difference in SOC stocks between the LSs and the ASs can be explained by that the LSs had lower SOC contents but deeper profiles (Fig. 1 and 2). Precipitation had significant effects on SOC stocks of the LSs, as indicated by the wet-LSs having greater SOC stocks than the dry-LSs (Fig. 2). This is consistent with the consensus that SOC stocks generally increase with

precipitation (Homann et al., 2007; Wiesmeier et al., 2019). The higher SOC stocks in the wet-LSs can be also explained by SOC contents because of (1) similar soil depths in the wet-LSs and the dry-LSs (Fig. 1) and (2) lower soil bulk densities in the wet-LSs (Table S2). Hence, patterns of SOC stocks controlled by lithology and precipitation are mainly explained by SOC contents.

The negative correlations between SOC contents and SOC mineralization rates (Fig. 7A and 7B) reflect

SOC contents controlled by SOC stability. The SOC stability is significantly controlled by precipitation and lithology (Table 1) rather than soil horizon, aggregate size or aggregate destruction (Fig. 6). For horizons, although SOC stability was different between A and B horizons in the crushed SM fraction of the dry-LSs (Table 1), the small contribution of the SM fraction (Fig. 3) suggest that horizon is not an important factor controlling the SOC stability.


### 4.3 Organic matter stabilization mechanisms

SOC stability is largely controlled by two mechanisms: (1) OM adsorption on the mineral surfaces and (2) physical occlusion of OM within soil aggregates (Lützow et al., 2006; Six et al., 2002). Overall, aggregate-occlusion is not a major OM stabilization mechanism in these soils, as indicated by no or

insignificant stimulation in SOC mineralization after aggregate destruction (Fig. 6). The minor role of aggregate-occlusion is further supported by the minor changes in correlation patterns of SOC mineralization rates with SOC contents and C/N ratios, when aggregates were intact and crushed (Fig. 7A and 7B). The limited effects of OM occlusion in aggregates are not consistent with the general view of aggregate-controlled OM stabilization (Lehmann and Kleber, 2015; Wiesmeier et al., 2019), as well as

other studies revealing aggregate-protected OM using similar aggregate destruction methods (Mueller et al., 2012; Wang et al., 2014). However, Goebel *et al.* (2009) and Juarez *et al.* (2013) reported limited roles of soil aggregates in protecting OM from decomposition. For the ASs, the limited role of aggregates



in OM stabilization can be explained by the lack of large-sized aggregates (Fig. 3), which suggest the restricted formation of microaggregates within macroaggregates. This potentially weakens the OM

protection controlled by occluded in aggregates (Six et al., 2002; Six and Paustian, 2014). For the LSs, the minor contribution of aggregates might be related to the strong adsorption of OM on less-saturated mineral surfaces (Yang *et al.*, submitted). Because of the limited contribution of OM occlusion in aggregates, OM adsorption on mineral surfaces is most likely the dominant stabilization mechanism. Similar to our results, mineral-controlled OM stabilization mechanisms have been reported in other

studies in alpine grassland soils of the Andes (Yang *et al.,* submitted; Buytaert *et al.*, 2006a; Tonneijck *et al.*, 2010; Rolando *et al.*, 2017b).

Lithology is an important factor for OM stabilization related to mineral surfaces. Yang *et al.* (submitted) found that OM stabilization in the wet-LSs was controlled by OM complexed and/or adsorbed with Fe and Al (oxides) as well as by $Ca^{2+}$ bridges. In contrast, OM stabilization in the wet-ASs was only

controlled by Fe and Al (oxides) complexation (Yang *et al.* submitted). In the wet site, SOC stability between LSs and ASs was not significantly different (Table 1). This may be attributed to the mineral surfaces in both LSs and ASs having a large capability for OM stabilization, although their OM stabilization mechanisms are slightly different. In the dry site, lower SOC stability in the LSs compared to the ASs (Table 1) suggests the lower capacity of the mineral surfaces to stabilize OM in the LSs.

Similarly, Heckman *et al.* (2009) found lower SOC stocks and stability in LSs compared to soils formed on felsic and basaltic igneous rocks, in a region with similar temperature and precipitation to our dry site. They explained this by a lack of active Fe and Al fractions to stabilize OM (Heckman et al., 2009), which might be an explanation for the less stable SOC in our dry-LSs as well.

Precipitation is also an important factor to explain the low SOC stability in the dry-LSs, as precipitation

has a potential effect on soil mineralogy by controlling weathering processes (Doetterl et al., 2015, 2018; Wiesmeier et al., 2019). Compared to the wet-LSs, the lower pH values in the dry-LSs indicate that a part of exchangeable base cations (e.g $Ca^{2+}$) are replaced by exchangeable $H^+$. The replacement results in lower adsorption capacity of the mineral surfaces because $H^+$ is a monovalent cation that does not promote OM stabilization (Jenny, 1994; Lützow et al., 2006).  In addition, positive correlations between

SOC mineralization rates and SOC contents, and between SOC mineralization and C/N ratios in the dry-LSs (Fig. 7) indicate that SOC mineralization is dominantly dependent on SOC contents and quality. This also suggests a lower sorption capacity of the mineral adsorption sites. Similarly, Wagai *et al.* (2008) reported positive correlations between SOC mineralization and C/N ratios, and used the positive correlations as an indication of inert mineral surfaces. Furthermore, the lowest C/N ratios in the dry-LSs





(Fig. 2) indicate a depletion of plant-derived C and a rapid SOC decomposition process (Moni et al.,

2012), which suggest the low SOC stability and the low capacity of mineral surfaces to stabilize OM.

*4.4 Interactions between precipitation and lithology*

The effects of precipitation and lithology on SOC stocks and stability are unlikely through the controls of

soil aggregation, which is supported by the weak controls of OM stabilization via occlusion in aggregates

(Fig. 6 and 7) and inconsistent patterns of aggregate size distribution compared to the patterns of SOC

stability (Fig. 3, Fig. 4 and Table 1). In contrast, the interactions between precipitation and lithology on

SOC stocks and stability are likely explained by soil mineralogy. This is supported by (1) the contrasting

OM stabilization mechanisms controlled mineral surfaces between the wet-LS and the wet-AS (Yang *et*

*al.*, submitted), and (2) shifts in pH values, C/N ratios and correlations between SOC mineralization rates

and SOC contents that suggest variations in properties of the mineral surfaces (Fig. 2 and Fig. 7).

Recent studies indicate that controls of climate factors and soil mineralogy are crucial to the persistence

and stabilization of soil OM (Chaplot et al., 2010; Doetterl et al., 2015; Homann et al., 2007). For the LSs,

we proposed that the lower SOC stability in the dry site is explained by the weaker interactions between

OM and mineral surfaces due to the lower pH when compared to the wet site. However, the lower pH in

the dry-LSs is not consistent with the general soil formation process. The lower pH in the dry-LSs might

be explained by soil acidification induced by higher below-ground OM input compared to the wet-LSs.

The higher below-ground input is indicated by more α, ω-dioic acids, ω-hydroxyl alkanoic acids and

long-chain fatty acids (C20-32) in the dry-LSs when compared to the wet-LS, especially in B horizons

(Fig. S2), because these compounds are mainly derived from root input (Kögel-Knabner, 2002). The

higher and deeper OM input in the dry-LSs can be explained by the lower precipitation, for which plants

need more developed root systems. By contrast, no clear difference is found between the wet-ASs and the

dry-ASs (Fig. 1 and Table 1). This may be attributed to the limited acidification induced by OM input

because the bedrocks are already acidic.

Similar to our results, Wagai et al. (2008) reported that the controls of altitude (temperature and

precipitation) on OM stoichiometry (indicating mineral surface activity) are dependent on soil bedrocks.

Furthermore, Doetterl et al. (2015, 2018) indicated that climate factors in relation to soil mineralogy

control the potential of soil matrix to stabilize OM. Our findings also support their views that the OM

persistence is controlled by climate factors and soil mineralogy. We further propose that the interactions



between precipitation and lithology on OM stabilization in our study are through the controls of soil
mineralogy in relation to OM input.

## 5 Conclusion

Our findings highlighted (1) SOC stocks and stability controlled by interactions between precipitation and
lithology, and (2) soil aggregate size distribution controlled by lithology only. As the assumption that
aggregate occlusion contributes to OM stabilization is not supported by our data, we conclude that OM
adsorption on mineral surfaces is the major OM stabilization mechanism in these soils. We propose that
the controls of precipitation and lithology on SOC stocks and OM stabilization are through the controls of
soil mineralogy in relation to OM input.

Further studies are required for more lithology types and more precipitation levels. In addition, primary
effects of precipitation on OM dynamics are not limited to the controls of soil mineralogy. Potential
effects of precipitation on quantity and quality of input OM suggest that investigations in OM molecular
composition may contribute to a better understanding of the processes governing SOC sequestration in
the Neotropical grasslands of the Andes.


**Author contribution.** SY, BJ, KK and EC conceived and designed the study; RvH contributed to the
experiments related to aggregate-size fractionation and analyses of soil properties; SA contributed to the
soil incubation and the SOC mineralization measurement; SY wrote the paper. All authors contributed to
the manuscript revision.


**Competing interests.** The authors declare that they have no conflict of interest.

**Acknowledgement.** We thank Lisa Boerdam, Chiara Cerli and Eva de Rijke for their help in lab work, as
well as Xiang Wang for sharing his experiences for the incubation experiment. We thank Fresia Olinda
Chunga Castro for her assistant in the field sampling. We also thank the Mountain Institute (TMI) for
their support in the field work, and Institute for Biodiversity and Ecosystem Dynamics (IBED) and China
Scholarship Council (CSC) for funding.



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

Lithology controlled soil organic carbon stabilization in an alpine grassland of the Peruvian Andes,
Submitt to Environ. Earth Sci.



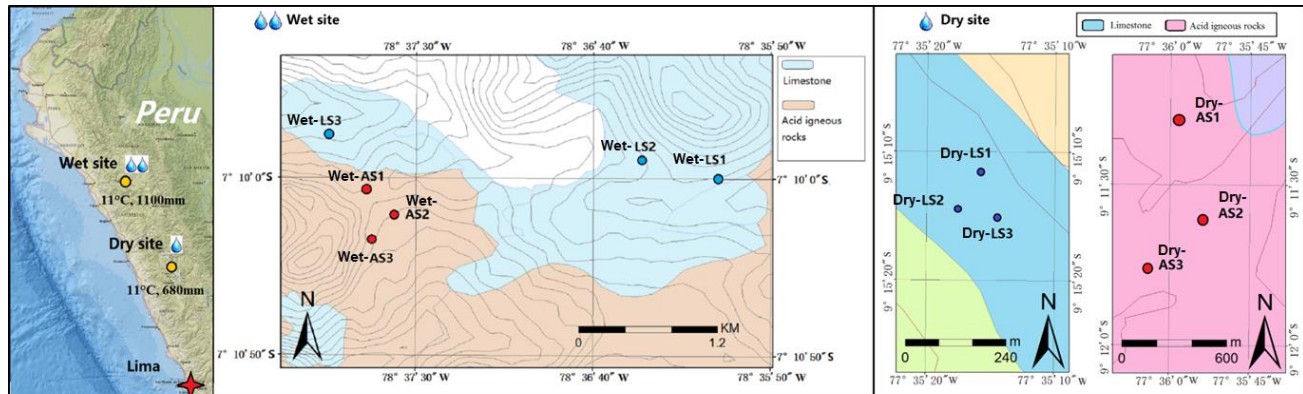

| Site | Wet site | | | | | Dry site | | | | |
| | Parent material | Altitude | Soil depth | Ave. depth | Gravels in LM fractions | Parent material | Altitude | Soil depth | Ave. depth | Gravels in LM fractions |
| --- | --- | --- | --- | --- | --- | --- | --- | --- | --- | --- |
| | | m | cm | | % | | m | cm | | % |
| LS1 | Limestone | 3716 | 57 | | 6.86 | Limestone | 3573 | 56 | | 8.70 |
| LS2 | Limestone | 3717 | 66 | 61 | 3.13 | Limestone | 3532 | 54 | 61 | 5.42 |
| LS3 | Limestone | 3517 | 60 | | 0.16 | Limestone | 3560 | 73 | | 13.01 |
| AS1 | Granite/ignimbrite | 3583 | 68 | | 10.86 | Granodiorite | 3667 | 44 | | 50.28 |
| AS2 | Granite/ignimbrite | 3585 | 45 | 49 | 14.12 | Granodiorite-rich glacier materials | 3521 | 60 | 51 | 39.57 |
| AS3 | Granite/ignimbrite | 3586 | 35 | | 28.60 | Granodiorite-rich glacier materials | 3495 | 50 | | 61.05 |

**Fig. 1 Sampling site description.** LS: limestone soil, AS: acid igneous rock soil, LM: large macroaggregates (>2 mm). The ArcGIS Online World Topographic Map basemap (Esri., 2013) was used for the map of Peru on the left, whereas the data for the contour lines in the maps of the wet site and the dry site was derived from Geo GPS Perú, (2014).



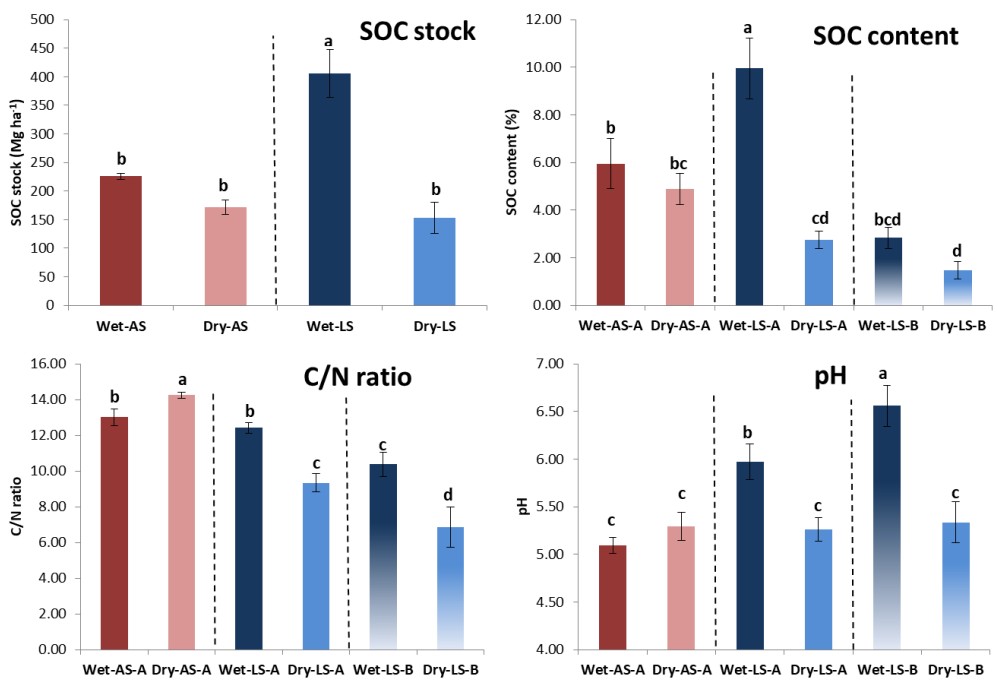

**Fig. 2 Soil organic carbon stocks in the whole soil profile and soil properties in diagnostic horizons (Mean±SE).** Wet: the wet site, Dry: the dry site, LS: limestone soil, AS: acid igneous rock soil, A: A horizons, B: B horizons

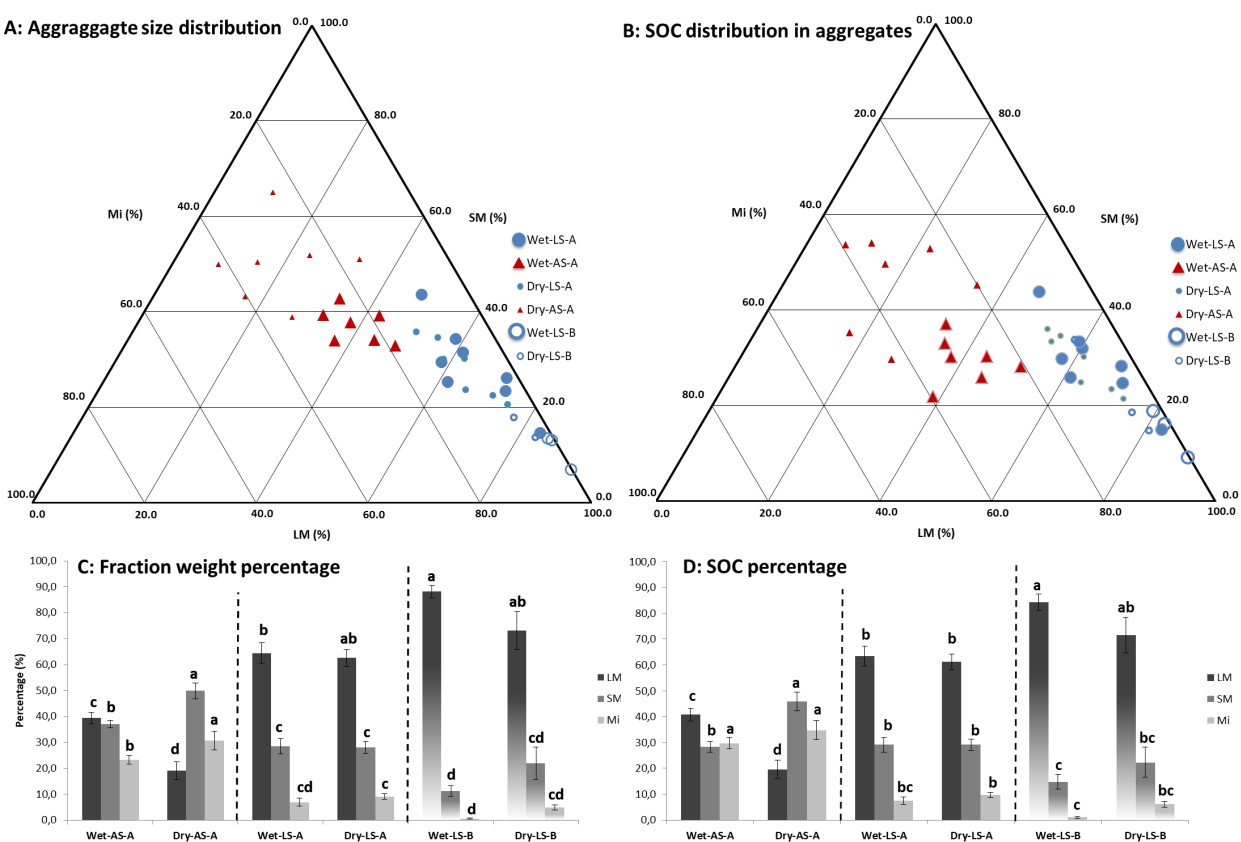

**Fig. 3 Distribution of fraction weight and soil organic carbon in aggregate size fractions.** A: Fraction weight distribution in aggregate size fractions, B: SOC distribution in aggregate size fractions, C: percentages of fraction weights in soil horizons (Mean±SE), D: SOC percentage in soil horizons (Mean±SE). Wet: the wet site, Dry: the dry site, LS: limestone soil, AS: acid igneous rock soil, A: A horizons, B: B horizons, LM: large macroaggregates (>2 mm), SM: small macroaggregates (0.25-2 mm), Mi: microaggreagtes (<0.25 mm)



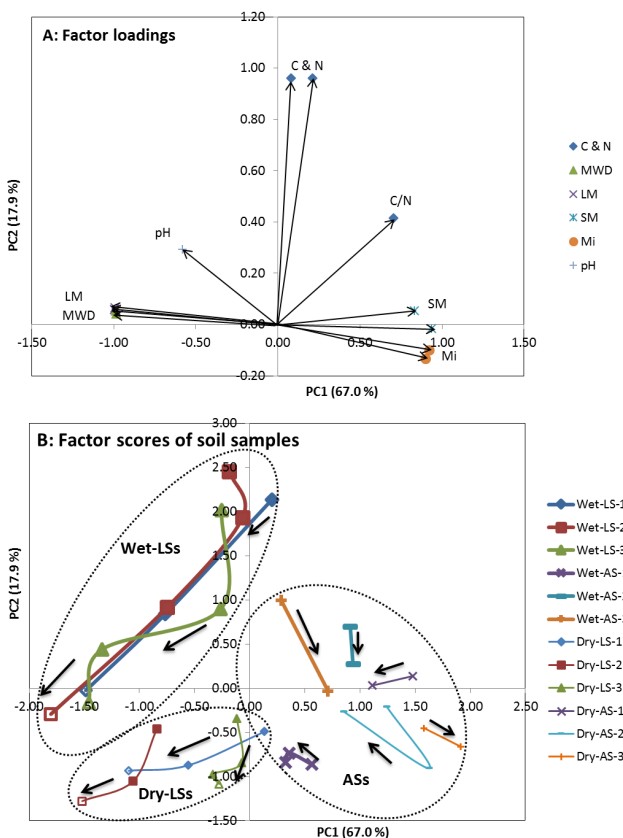

**Fig. 4 Principal component analysis (PCA) indicating vertical distribution of aggregate-related soil properties in both limestone soils (LSs) and acid igneous rock soils (ASs).** Solid points are A horizons, and hollow points are B horizons. Black arrows are pointing to the direction of soil horizons with increasing soil depth. Wet: the wet site, Dry: the dry site, LS: limestone soil, AS: acid igneous rock soil, MWD: mean weight diameter, C: SOC content, N: total nitrogen content, C/N: C/N ratio, LM: large macroaggregates, SM: small macroaggregates, Mi: microaggregates.



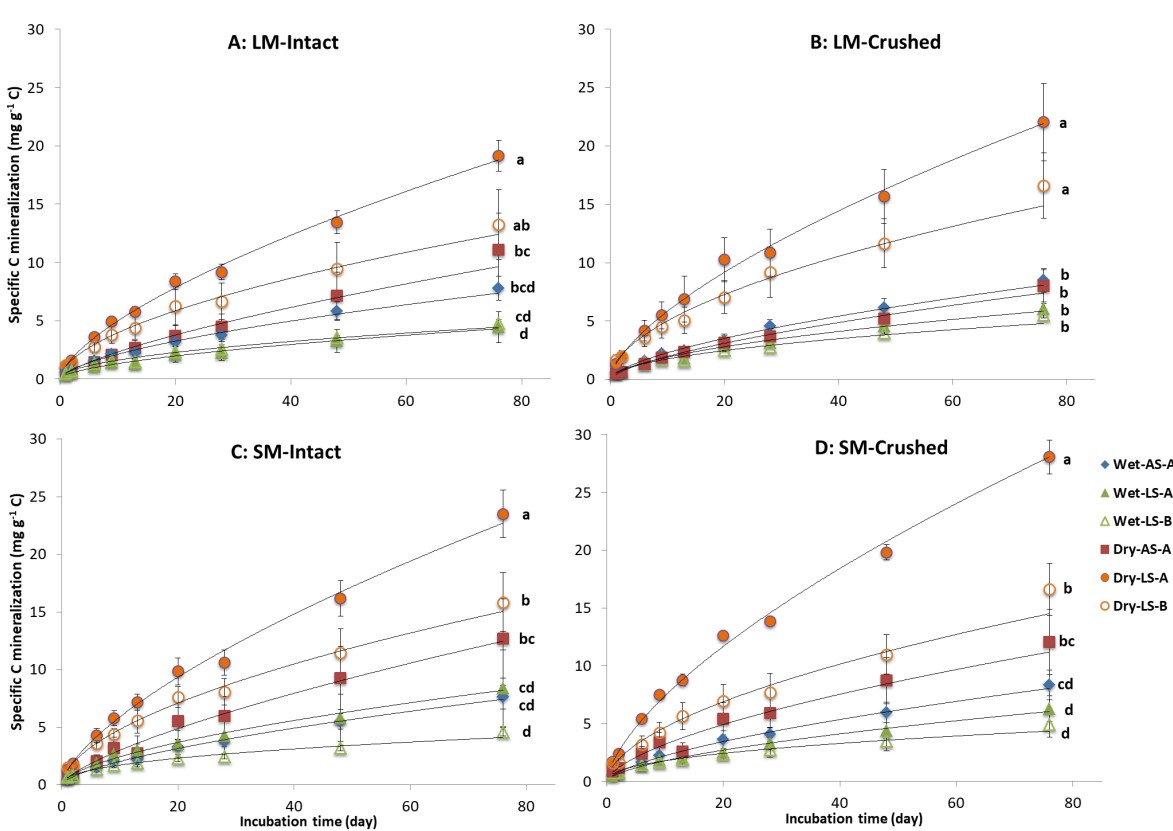

**Fig. 5 SOC mineralization in the large macroaggregats (LM) and small macroaggregates (SM) in a period of 76-day incubation, with aggregate intact and crushed (Mean ± SE).** Letters on the right of each plots indicate significant differences of cumulative C mineralization between different groups on Day 76. LM: large macroaggregates (>2 mm), SM: small macroaggregates (0.25-2 mm), Intact: incubation with aggregates intact, Crushed: incubation with aggregates crushed, Wet: the wet site, Dry: the dry site, AS-A: acid igneous rock soil - A horizon, LS-A: limestone soil - A horizon, LS-B: limestone soil - B horizon.

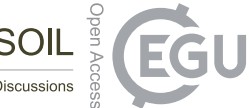

**Table 1 Comparison in SOC mineralization rates between bedrock, precipitation and horizon with combinations of aggregate sizes and aggregate destruction.** Abbreviations in the table indicating the group with significant higher SOC mineralization than the other group.

| | A horizon: LS vs. AS | | | | A horizon: Wet vs. Dry | | | | LS: A vs. B horizon | | | |
|---|---|---|---|---|---|---|---|---|---|---|---|---|
| | LM-In | LM-Cr | SM-In | SM-Cr | LM-In | LM-Cr | SM-In | SM-Cr | LM-In | LM-Cr | SM-In | SM-Cr |
| | **Wet** | | | | **LS** | | | | **Wet** | | | |
| **Day 1** | n.s. | n.s. | n.s. | n.s. | Dry** | Dry* | Dry* | Dry** | n.s. | n.s. | n.s. | n.s. |
| **Day 2** | n.s. | n.s. | n.s. | n.s. | Dry** | Dry* | n.s. | Dry** | n.s. | n.s. | n.s. | n.s. |
| **Day 6** | n.s. | n.s. | n.s. | n.s. | Dry* | n.s. | n.s. | Dry** | n.s. | n.s. | n.s. | n.s. |
| **Day 9** | n.s. | n.s. | n.s. | n.s. | Dry** | n.s. | n.s. | Dry** | n.s. | n.s. | n.s. | n.s. |
| **Day 13** | n.s. | n.s. | n.s. | n.s. | Dry* | n.s. | Dry* | Dry** | n.s. | n.s. | n.s. | n.s. |
| **Day 20** | n.s. | n.s. | n.s. | n.s. | Dry* | n.s. | Dry* | Dry** | n.s. | n.s. | n.s. | n.s. |
| **Day 28** | n.s. | AS* | n.s. | n.s. | Dry** | n.s. | Dry* | Dry** | n.s. | n.s. | n.s. | n.s. |
| **Day 48** | n.s. | n.s. | n.s. | n.s. | Dry** | Dry* | Dry* | Dry** | n.s. | n.s. | n.s. | n.s. |
| **Day 76** | n.s. | n.s. | n.s. | n.s. | Dry** | Dry* | Dry* | Dry** | n.s. | n.s. | n.s. | n.s. |
| | **Dry** | | | | **AS** | | | | **Dry** | | | |
| **Day 1** | LS* | n.s. | LS* | n.s. | n.s. | n.s. | n.s. | n.s. | n.s. | n.s. | n.s. | n.s. |
| **Day 2** | LS* | LS* | LS* | n.s. | n.s. | n.s. | n.s. | n.s. | n.s. | n.s. | n.s. | n.s. |
| **Day 6** | LS* | n.s. | n.s. | LS* | n.s. | n.s. | n.s. | n.s. | n.s. | n.s. | n.s. | A* |
| **Day 9** | LS* | LS* | n.s. | LS* | n.s. | n.s. | n.s. | n.s. | n.s. | n.s. | n.s. | A* |
| **Day 13** | n.s. | n.s. | LS* | LS** | n.s. | n.s. | n.s. | n.s. | n.s. | n.s. | n.s. | A** |
| **Day 20** | n.s. | LS* | n.s. | LS* | n.s. | n.s. | n.s. | n.s. | n.s. | n.s. | n.s. | A* |
| **Day 28** | n.s. | n.s. | n.s. | LS** | n.s. | n.s. | n.s. | n.s. | n.s. | n.s. | n.s. | A** |
| **Day 48** | n.s. | LS* | n.s. | LS** | n.s. | n.s. | n.s. | n.s. | n.s. | n.s. | n.s. | A** |
| **Day 76** | n.s. | LS* | n.s. | LS** | n.s. | n.s. | n.s. | n.s. | n.s. | n.s. | n.s. | A** |

LS: limestone soil, AS: acid igneous rock soil, LM: large macroaggregates (>2 mm), SM: small macroaggregates (0.25-2 mm), MA: macroaggregates (>0.25 mm), A: A horizon, In: aggregate intact, Cr: aggregate crushed, Wet: the wet site, Dry: the dry site, *: P<0.05 **: P<0.01, n.s.: not significant.

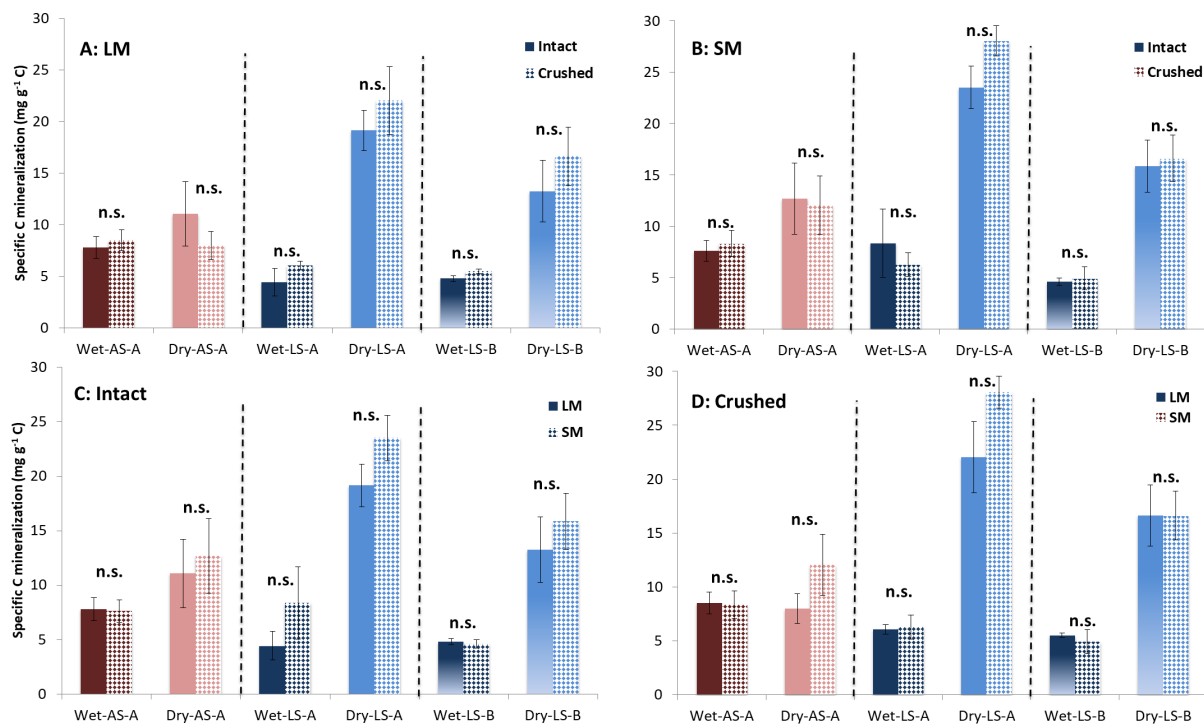

**Fig. 6 Effects of aggregate destruction and aggregate size on specific SOC mineralization rates in the sampling day 76 (Mean ± SE).** A: comparing aggregates intact and crushed in large macroaggregates, B: comparing aggregate intact and crushed in small macroaggregates, C: comparing large and small aggregates with aggregates intact, D: comparing large and small aggregates with aggregates crushed. LS: limestone soil, AS: acid igneous rock soil, LM: large macroaggregates (>2 mm), SM: small macroaggregates (0.25-2 mm), Intact: incubation with aggregates intact, Crushed: incubation with aggregates crushed, A: A horizon, Wet: the wet site, Dry: the dry site, n.s.: not significant.



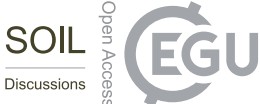

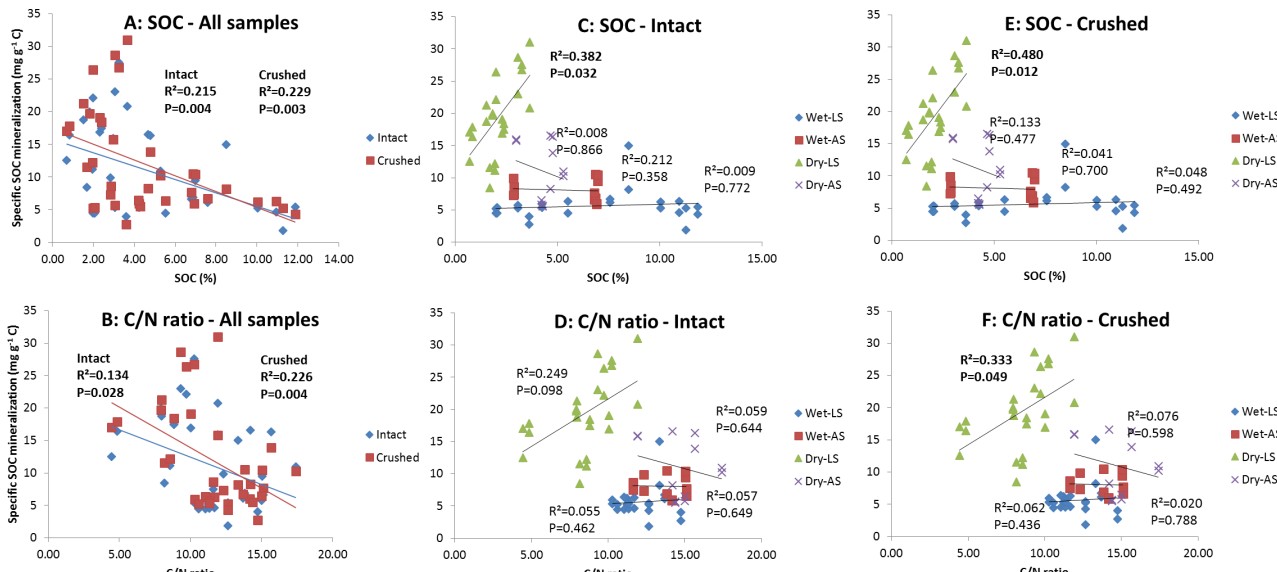

**Fig. 7 Relationships of specific C mineralization rates (Day 76) with organic carbon contents and C/N ratios when soil aggregates were intact and crushed.** Wet: the wet site, Dry: the dry site, LS: limestone soil, AS: acid igneous rock soils, SOC: soil organic carbon content.