# Peer review of "Lithology and climate controlled soil aggregate size distribution and organic carbon stability in the Peruvian Andes"

_SOIL, 2019_

## Referee Comment (RC1) · Anonymous Referee #1 · 22 Aug 2019

Soil organic matter storage and sequestration is a function of multiple factors from climate over parent material to plant traits and soil biology. The experimental differentiation between these factors in order to assess what drives SOC sequestration at a given soil ecosystem is always a tricky thing as it includes often multiple assumptions and or possible artefacts. The authors were up for the endeavour and aimed to elucidate how lithology and precipitation affect SOC stability in the Peruvian Andes. On top of this, the authors tried to measure if aggregation is the key factor that determines SOC stability at the study sites. This work in general is of special interest to a wider community working on the stabilization and destabilization of SOC. As the authors used a manipulation of the aggregation of the soils to compare intact and disrupted soils, this study

adds a novel aspect to an ongoing discussion on the importance of aggregation vs. mineral-association. Furthermore, there are not so many studies that tried this so far. Thus, the study is well suited for SOIL and also might provide some new insights into SOC sequestration in soils of the Peruvian Andes. Nevertheless, I think there are some major points that should be addressed by the authors, namely vegetation and plant trait information, incubation conditions of crushed vs. intact soils in terms of physical soil properties ($O_2$ diffusion, water content etc.) and a possible lack in temporal resolution due to the late start of the $CO_2$ measurements. All of these aspects are mentioned in the detailed remarks below and also commented with respect to possible improvements of the manuscript: line 20 and general - To improve readability I would use no abbreviations for the soils, there is enough space to use e.g. "limestone soils" instead of LS. line 30 How much occluded OM was present in these soils? Thus is aggregation at all relevant for OM storage in these soils in contrast to mineral association? line 40 The Andes stretch over 7000 km, please be more specific on the location of your work. line 44-47 Again, the Andes stretch over vast distances, its clear that there are drastic climatic differences. But even at one location you get changes with exposition and elevation. Please be more specific! line 50-51 This is redundant in itself, OM and OC is stabilized and of course is tightly linked. line 54-56 Again, this is highly dependent where you are in the Andes. In the Southern Andes you'll have soils that are completely dominated by particulate OM rather than mineral-associated OM. line 93 describe shortly Puna and Jalca line 103 Are there records about a longer consistent land use at the sampling sites? Or do you have indices that show a longer sustained land use type? line 122 So there was a mixture of different land uses between the three site replicates? Was this detectable in the soil profiles or SOM properties? line 147 How was the gravel content calculated, thus how did the authors differentiate between large aggregates (>2mm) and stones of this size range? line 157-158 Why you analyse in the one approach the fraction <63 $\mu$m, but don't use it in the incubation? Please describe here. line 162 By this approach you are not only crushing aggregates, but also rock fragments. How did you account for the different content of pure mineral

constituents in relation to aggregates? line 164 It was shown before that aggregate/soil disruption can lead to a rather fast spike in $CO_2$ evolution within a few days. Did you in any way account for this $CO_2$ loss between the different treatments during the first days of incubation before entering a sort of basal respiration? line 208-215 This is a nice exemplary paragraph to show how hard a text is to read for an outsider - let me summarize: "...LS is larger than AS, has more LM but minor Mi; AS has larger SM and Mi; LS were not different but wet-AS was slightly different from dry-AS..." I would really appreciate if you find a way to use even short words that are more descriptive and don't ruin the flow of reading. line 226-227 Please also give mineralization rates normalized to the amount of OC in the individual samples. This will give a better mechanistic insight on the fate of OM with respect to aggregation. This might also level off possible differences in stone content etc. line 233-234 How is this relation if you normalize OC mineralization rates with sample amount OC? line 239 Also if normalized on the amount of OC in LM vs. SM? line 253-256 You are using two very contrasting parent materials which foster completely different soil biological communities and soil chemistry and thus of course yield different soil structure - so far its textbook knowledge. Such statement might be more interesting if comparing Granodiorite and a Granite or Basalt etc. However, this comment is just about leaving out such "general textbook statements" and focus on the core of the story. line 256-260 This could possibly find its way into the Introduction as you could put this as a rational to take these two contrasting materials. In the discussion it appears again as a redundant textbook message. line 262 So basically the lack of fine material causes the lack of a more advanced aggregation. line 271 You are comparing a silicate rock and a carbonate rock - I would be more than surprised if precipitation would not have a less pronounced effect. line 275-280 There are in parts differences in aggregation and SOC stocks between wet and dry sites. Why are you neglecting those and talking them down as minor or biased by stoniness? If stoniness is the driving property, than how can you compare aggregate mineralization etc. at all? line 282 Given the high amount of stones and a some other constraints, the significant effects are worth taking them serious. Presumably as a result of altered soil biology and/or plant diversity / litter/root input. line 294-295 Which is a function of primary production and decomposition. Please give in the M&M more details on vegetation at the respective sites. line 299-304 What soil horizons comprise the low SOC values with high $CO_2$ evolution? Are those the low C/N ratio subsoils? If so, you are mixing two opposite factors, aggregation and soil material origin. Please give specific OC mineralization normalized per amount OC. And the very low C/N ratios under 5, would mean you have pure amino acid material in the sample. Could here values around the detection limit for N play a role? line 310-314 How much OM is stored within the aggregates? Do you have estimates of amounts of e.g. occluded POM? line 315 The cited work showed a clear effect of aggregate disruption within the first days of incubation. You lack this information due to the late start after 10 days. So the low differences between crushed and intact might be due to fact that you missed the $CO_2$ spike. Furthermore, how did you adjust comparable soil porosity/$O_2$ diffusion and thus water contents between finely crushed/ground soil material and naturally aggregated soil? line 343 Do you have data on exchangeable ions? line 358 How is the vegetation at the sites, how is primary production, above and belowground OM input? The biggest control on SOC stocks besides soil properties are plant traits at comparable parent materials. So as stated above, please give information on vegetation data in M&M. line 368-370 Or these compounds are just more stable at dry conditions. On top of that, plants produce e.g. more suberin in the roots as protection against drought. And without a baseline of the initial plant material above and belowground this data just tells you there are differences in these acids due to precipitation. line 376-377 There is the same amount of work showing plant species and traits having these effects on SOC storage and stability. Thus to prove the solely precipitation effect you would have to work with comparable plant species and traits. line 381 So how high is the OM input? line 385 You compare limestone with granodiorite, as mentioned above this of course outcompetes any effect of precipitation at same altitude and latitude. line 385-387 For this you would have to show that there is no occluded light fraction/POM, and you didn't miss the fast pulses (>10 days) in $CO_2$ after soil structure disruption found by others.

---

## Author Comment (AC1) · 13 Sep 2019

Dear Reviewer:

Thank you for your comments on this manuscript. We tried to address each point and answer your questions one by one. All Figures and Tables can be found at the end of this file.

**Here are the answers to the questions:**

**Line 20 and general** - To improve readability I would use no abbreviations for the soils, there is enough space to use e.g. "limestone soils" instead of LS.

**Answer:** Yes. We will correct them in the abstract.

**Line 30** How much occluded OM was present in these soils? Thus is aggregation at all relevant for OM storage in these soils in contrast to mineral association?

**Answer:** We have no idea of the amount of occluded POM because the POM cannot be separated using density fraction in the ASs. When sonication was used for the ASs, organic material was suspended in the Na polytungstate solution (1.6 g cm$^{-3}$). We tried many ways (including long-time centrifuge) to separate the organic materials but always failed. Thus, we had to change the way (i.e. incubation with aggregate intact and crushed) to estimate the OM occluded in aggregates. This way is indirect but still widely used to estimate OM protected by aggregates. As we found no difference in SOC mineralization between intact and crushed aggregates, therefore we proposed that occlusion in aggregates was not the major way that OM was stabilized in our soils.

**Line** 40 The Andes stretch over 7000 km, please be more specific on the location of your work.

**Answer:** This is meant as a general statement as in general these functions are valid for the alpine grassland (paramo-jalca-puna system) from Venezuela all the way to Bolivia, as they provide the arid western coast of South America with water and are a biodiversity hotspot, and high SOC stocks.

However, to clarify the exact location, we will add a more detailed description of the field sites at the end of this paragraph.

**Line** 44-47 Again, the Andes stretch over vast distances, its clear that there are drastic climatic differences. But even at one location you get changes with exposition and elevation. Please be more specific!

**Answer:** We will add information on this by extending this line with "and in this study we focus on two contrasting sites in the Andes in Northern Peru".

**Line** 50-51 This is redundant in itself, OM and OC is stabilized and of course is tightly linked.

**Answer:** This will be corrected.

**Line** 54-56 Again, this is highly dependent where you are in the Andes. In the Southern Andes you0ll have soils that are completely dominated by particulate OM rather than mineral-associated OM.

**Answer:** We will specify this with: Peruvian and Ecuadorian Andes.

**Line** 93 describe shortly Puna and Jalca

**Answer:** We will add in the site description: Wet Puna or Jalca is present between the tree line (3500 m asl) and the ice-covered region, having precipitation above 500 mm. For more ore information the reader is referred to the literature and the following contents in the M&M.

**Line** 103 Are there records about a longer consistent land use at the sampling sites? Or do you have indices that show a longer sustained land use type?

**Answer:** Please see our answer to Line 122.

**Line** 122 So there was a mixture of different land uses between the three site replicates? Was this detectable in the soil profiles or SOM properties?

**Answer:** Yes, samples were collected from grassland and abandoned cropland. A previous study showed that SOC stocks were not clearly affected by land use type (e.g. grassland vs. cultivation). This is because the local farmers rotate the land in the order of cultivation, abandoned cultivation, cultivate grassland and grassland in a period of several years. They repeated this cycle. This may keep the SOC stock high and in a dynamic balance. (Yang et al. 2018, Catena). We will modify the text to make this clearer.

**Line** 147 How was the gravel content calculated, thus how did the authors differentiate between large aggregates (>2mm) and stones of this size range?

**Answer:** The fraction larger than 2mm was obtained by sieving and stones were picked out by hand. From the remaining aggregates the fractions >5 mm and 2-5 mm were obtained after the dry sieving.

**Line** 157-158 Why you analyse in the one approach the fraction <63 _m, but don0t use it in the incubation? Please describe here.

**Answer:** The finer fraction (<0.25 mm) were by far less abundant, especially for the limestone soils. The fraction was therefore not incorporated in the analysis.

**Line** 162 By this approach you are not only crushing aggregates, but also rock fragments. How did you account for the different content of pure mineral constituents in relation to aggregates?

**Answer:** We will replace the word "grinding" with "crushing" here, as this is actually what we did.  If rock fragments are defined as size >2mm, this approach will not break rock fragments because rock fragments were removed before the aggregate crushing. We cannot account for the differences between mineral particles and aggregates. Nevertheless, our purpose was only crushing aggregates. Grinding using a porcelain mortar is unlikely to destruct mineral particles.

**Line** 164 It was shown before that aggregate/soil disruption can lead to a rather fast spike in $CO_2$ evolution within a few days. Did you in any way account for this $CO_2$ loss between the different treatments during the first days of incubation before entering a sort of basal respiration?

**Answer:** Thank you for this question. We applied a 10-day pre-incubation, necessary because microbes in the air-dried soils need to be activated. We used slow wetting because we wanted to avoid aggregate destruction caused by fast-wetting.

For the spike in $CO_2$ at the beginning, data from Fig. R1 indicated that the fast pulse in $CO_2$ was not missed. This is because soils were too dry for microbes to start degradation during the early period of the pre-incubation. Furthermore, the comparisons in Table R1 showed no clear differences in early $CO_2$ production between intact and crushed aggregates. Thus, our observation did not miss the massive $CO_2$ during the first days, whereas our results do not support that aggregate/soil disruption caused a fast spike in $CO_2$ evolution at the beginning.

**Line** 208-215 This is a nice exemplary paragraph to show how hard a text is to read for an outsider - let me summarize: "...LS is larger than AS, has more LM but minor Mi; AS has larger SM and Mi; LS were not different but wet-AS was slightly different from dry-AS..." I would really appreciate if you find a way to use even short words that are more descriptive and don0t ruin the flow of reading.

**Answer:** Thank you for this point. We will try to improve the wording if the manuscript if we are allowed to revise.

**Line** 226-227 Please also give mineralization rates normalized to the amount of OC in the individual samples. This will give a better mechanistic insight on the fate of OM with respect to aggregation. This might also level off possible differences in stone content etc.

**Answer:** SOC mineralization was normalized for OC contents already. We will add a line on this to the methods section: SOC mineralization rates provided were normalized for OC contents.

**Line** 233-234 How is this relation if you normalize OC mineralization rates with sample amount OC?

Answer: see our previous answer to the question Line 226-227.

**Line** 239 Also if normalized on the amount of OC in LM vs. SM?

**Answer:** See our previous answer to the question Line 226-227.

**Line** 253-256 You are using two very contrasting parent materials which foster completely different soil biological communities and soil chemistry and thus of course yield different soil structure - so far its textbook knowledge. Such statement might be more interesting if comparing Granodiorite and a Granite or Basalt etc. However, this comment is just about leaving out such "general textbook statements" and focus on the core of the story.

**Answer:** The statements concerning the effects of lithology on soil aggregate size distribution will be largely removed from the discussion part and moved, in a modified form, to the introduction.

**Line** 256-260 This could possibly find its way into the Introduction as you could put this as a rational to take these two contrasting materials. In the discussion it appears again as a redundant textbook message.

**Answer:** The contents will be rephrased and moved to the Introduction.

**Line** 262 So basically the lack of fine material causes the lack of a more advanced aggregation.

**Answer:** This is correct.

**Line** 271 You are comparing a silicate rock and a carbonate rock - I would be more than surprised if precipitation would not have a less pronounced effect.

**Answer:** The contents will be rephrased and moved to the Introduction.

**Line** 275- 280 There are in parts differences in aggregation and SOC stocks between wet and dry sites. Why are you neglecting those and talking them down as minor or biased by stoniness? If stoniness is the driving property, than how can you compare aggregate mineralization etc. at all?

**Answer:** We agree that there are major differences in aggregation and slight (non-significant) differences in SOC stocks. However, it is difficult to explain the differences in stoniness, as we found properties of each aggregate fractions were similar between wet-ASs and dry-AS. However, we could elaborate on the effects of differences in stoniness: root distribution will be different in stones, differences in soil moisture redistribution affecting soil microbial activity and organic matter turnover. We will more explicitly discuss the differences in aggregation between the two sites in a revised version of the manuscript.

**Line** 282 Given the high amount of stones and a some other constraints, the significant effects are worth taking them serious. Presumably as a result of altered soil biology and/or plant diversity / litter/root input.

**Answer:** Please see the answer to the previous question.

**Line** 294-295 Which is a function of primary production and decomposition. Please give in the M&M more details on vegetation at the respective sites.

**Answer:** Information on vegetation will be given in the M&M section:

The vegetation in the wet site is a typical disturbed wet Puna (or Jalca) vegetation with dominant grass species: *Calamagrostis sp.,* but also *Festuca and Agrostis sp. as well as Rumex sp. on fallow land.*

The vegetation in the dry site is a typical disturbed wet Puna (or Jalca) vegetation with *Calamogrostis sp., Stipa and Festuca sp. and Rumex sp. on fallow land.*

**Line** 299-304 What soil horizons comprise the low SOC values with high CO2 evolution? Are those the low C/N ratio subsoils? If so, you are mixing two opposite factors, aggregation and soil material origin. Please give specific OC mineralization normalized per amount OC. And the very low C/N ratios under 5, would mean you have pure amino acid material in the sample. Could here values around the detection limit for N play a role?

**Answer:** The mineralization rates were already normalized (Questions **Line** 226-227, **Line** 233-234 and **Line** 239) . In general, Dry-LS-A had the highest CO2 evolution, whereas Dry-LS-B had the lowest SOC contents. For the C/N ratios, the values were 9.34±0.52 for the Dry-LS-A and 6.86±1.14 for the Dry-LS-B (Fig. 2). Thus, soil horizons with the highest CO2 productions were not subsoils or the horizons with the lowest C/N ratios. In addition, we don't think the N contents reached the detection limit of the Elementar Analyzer because the detection limit of the Analyzer was 0.01% but the lowest N content was 0.16%.

**Line** 310-314 How much OM is stored within the aggregates? Do you have estimates of amounts of e.g. occluded POM?

**Answer:** See our answer to the question concerning line 30.

**Line** 315 The cited work showed a clear effect of aggregate disruption within the first days of incubation. You lack this information due to the late start after 10 days. So the low differences between crushed and intact might be due to fact that you missed the $CO_2$ spike. Furthermore, how did you adjust comparable soil porosity/$O_2$ diffusion and thus water contents between finely crushed/ground soil material and naturally aggregated soil?

**Answer:** We totally agree with that there is a fast spike in $CO_2$ at the beginning of the incubation. However, we had to re-wet the air-dried soils to initiate the decomposition. We choose to slowly re-wet soil materials for 10 days because fast-wetting can significantly break soil aggregates. We just would like to avoid unnecessary destruction of aggregates. At the first few days of the incubation, soil materials are very dry and the SOC mineralization did not start. Thus, the fast spike in $CO_2$ did not appear in this period.

Although we applied the pre-incubation, we believe that we did not miss the massive $CO_2$ production at the beginning. This is because of the much higher $CO_2$ production rates in the first few days of the measurement (Fig. R1). In many studies, the pre-incubations were 14 days. Luckily, we anticipated the fast spike in $CO_2$ at the beginning and we try to shorten the pre-incubation time. If we pre-incubated soils for 14 days as many studies did, we would be more likely missing the $CO_2$ spike that was found in Day 1 and Day 2 (Fig. R1).

For the adjustments of soil porosity and $O_2$ diffusion, we did not make them comparable for crushed vs. naturally aggregated soils. OM stabilization through occluded in aggregates can be explained by physical inaccessibility to the decomposer. The inaccessibility is closely related to the microstructure of aggregates (e.g. soil porosity and $O_2$ diffusion). The objectives of crushing aggregates were to destruct soil structure (i.e. soil porosity, $O_2$ diffusion, etc.) that promote OM stabilization. If we made soil porosity and $O_2$ diffusion comparable between intact and crushed aggregates, we were a bit like trying to eliminate what we want to compare.

**Line** 343 Do you have data on exchangeable ions?

**Answer:** We don't have the data of exchangeable ions for our sample sets. However, pH is an important factor for the ratio of exchangeable $Ca^{2+}$ and $H^+$. Thus, we use pH as an indicator. Nevertheless, we had

another paper (Yang, in revision, Envir. Earth Scie.) focusing on the effects of exchangeable ions, Fe and Al on SOM stabilization.

**Line** 358 How is the vegetation at the sites, how is primary production, above and belowground OM input? The biggest control on SOC stocks besides soil properties are plant traits at comparable parent materials. So as stated above, please give information on vegetation data in M&M.

**Answer:** Information on vegetation will be given in the M&M section:

The vegetation in the wet site is a typical disturbed wet Puna (or Jalca) vegetation with dominant grass species: *Calamagrostis sp.,* but also *Festuca and Agrostis sp. as well as Rumex sp. on fallow land.*
The vegetation in the dry site is a typical disturbed wet Puna (or Jalca) vegetation with *Calamogrostis sp., Stipa and Festuca sp. and Rumex sp. on fallow land.*
Based on the information, the vegetation is similar between the wet and the dry sites.

**Line** 368-370 Or these compounds are just more stable at dry conditions. On top of that, plants produce e.g. more suberin in the roots as protection against drought. And without a baseline of the initial plant material above and belowground this data just tells you there are differences in these acids due to precipitation.

**Answer:** For the first point whether these compounds are more stable in the dry site, our unpublished data showed that they are more vulnerable in the dry-LSs. This is evidenced from the Dry-LSs having a more clear trend in the depletion in α, ω-dioic acids and ω-hydroxyl alkanoic acids (maybe also long-chain fatty acids) than the Wet-LSs (Fig. R2). If these compounds are larger and meanwhile more vulnerable in the Dry-LSs compared to the Wet-LSs, a possible reason is that their input is higher. As it is very difficult to estimate OM input in the puna grassland, we can only assess these potential differences using the data of SOM composition.

**Line** 376-377 There is the same amount of work showing plant species and traits having these effects on SOC storage and stability. Thus to prove the solely precipitation effect you would have to work with comparable plant species and traits.

**Answer:** The vegetation between the two sites is slightly different, but consists of grasses of the same functional types and genera but with different (sub-)species. We speculate that their impact on the soil is comparable. With regard to the primary production we have no data and literature on this is also very scarce but we expect that NPP is also affected by the availability of moisture.

**Line** 381 So how high is the OM input?

**Answer:** We do not know the exact OM input as it is very difficult to estimate OM input in the Andean Puna/Jalca grassland. In addition, literature on OM production or NPP is very limited (one publication for Peru on slightly drier sites indicates a NPP of about 5 Mg C / ha yr for grazed grassland and around 15 Mg C /Ha yr (Oliveras et al. 2014, Environmental Research Letters), which might give an indication of the NPP at our sites. We will include this information in the manuscript.

**Line** 385 You compare limestone with granodiorite, as mentioned above this of course outcompetes any effect of precipitation at same altitude and latitude.

**Answer:** We will rephrase it: We did not find an important effect of precipitation on aggregation, which was overshadowed by the effect of different lithologies on aggregation

**Line** 385-387 For this you would have to show that there is no occluded light fraction/POM, and you didn0t miss the fast pulses (>10 days) in CO2 after soil structure disruption found by others.

**Answer:** For the occluded OM, as density fractionation was not applicable for the acid rock soils, incubation with aggregate intact vs. crushed is an alternative method to estimate occluded OM. For the fast pulses, data from Fig. R1 indicated that the fast pulse in CO2 was not missed, whereas the comparisons in Table R1 indicated that SOC production in the first days was not significantly higher for crushed aggregates than intact aggregates. Thus, we can propose that SOM is unlikely stabilized by occluded in aggregates. For details, please check the answers to the questions of **Line** 30, **Line** 164 and **Line** 315.

[Figure]

**Fig. R1 Specific SOC mineralization rate per day (g C mineralized g⁻¹ SOC day⁻¹).** Wet: the wet site, Dry: the dry site, LS: limestone soil, AS: acid igneous rock soil, A: A horizon, B: B horizon, LM: large macroaggregates (>2 mm), SM: small macroaggregates (0.25-2mm).

**Table R1 Comparisons in SOC mineralization rates (per day) between intact aggregates and crushed aggregates**

| | | Wet-LS-A | | Wet-LS-B | | Wet-AS-A | | Dry-LS-A | | Dry-LS-B | | Dry-AS-A | |
|---|---|---|---|---|---|---|---|---|---|---|---|---|---|
| | | LM | SM | LM | SM | LM | SM | LM | SM | LM | SM | LM | SM |
| Day1 | SMR | n.s. | n.s. | n.s. | n.s. | n.s. | n.s. | n.s. | n.s. | n.s. | n.s. | n.s. | n.s. |
| | SMR per day | n.s. | n.s. | n.s. | n.s. | n.s. | n.s. | n.s. | n.s. | n.s. | n.s. | n.s. | n.s. |
| Day2 | SMR | n.s. | n.s. | n.s. | n.s. | n.s. | n.s. | n.s. | n.s. | n.s. | n.s. | n.s. | n.s. |
| | SMR per day | n.s. | n.s. | n.s. | n.s. | n.s. | n.s. | n.s. | n.s. | n.s. | n.s. | n.s. | n.s. |
| Day6 | SMR | n.s. | n.s. | n.s. | n.s. | n.s. | n.s. | n.s. | n.s. | n.s. | n.s. | n.s. | n.s. |
| | SMR per day | n.s. | n.s. | n.s. | n.s. | n.s. | n.s. | n.s. | n.s. | n.s. | n.s. | n.s. | n.s. |
| Day9 | SMR | n.s. | n.s. | n.s. | n.s. | n.s. | n.s. | n.s. | n.s. | n.s. | n.s. | n.s. | n.s. |
| | SMR per day | n.s. | n.s. | n.s. | n.s. | n.s. | n.s. | n.s. | n.s. | n.s. | n.s. | n.s. | n.s. |
| Day13 | SMR | n.s. | n.s. | n.s. | n.s. | n.s. | n.s. | n.s. | n.s. | n.s. | n.s. | n.s. | n.s. |
| | SMR per day | n.s. | n.s. | n.s. | n.s. | n.s. | n.s. | n.s. | n.s. | n.s. | n.s. | n.s. | **In>Cr\*\*** |
| Day20 | SMR | n.s. | n.s. | n.s. | n.s. | n.s. | n.s. | n.s. | n.s. | n.s. | n.s. | n.s. | n.s. |
| | SMR per day | n.s. | n.s. | n.s. | n.s. | n.s. | n.s. | n.s. | n.s. | n.s. | n.s. | n.s. | n.s. |
| Day28 | SMR | n.s. | n.s. | n.s. | n.s. | n.s. | n.s. | n.s. | n.s. | n.s. | n.s. | n.s. | n.s. |
| | SMR per day | n.s. | n.s. | n.s. | n.s. | n.s. | n.s. | n.s. | n.s. | n.s. | n.s. | n.s. | n.s. |
| Day48 | SMR | n.s. | n.s. | n.s. | n.s. | n.s. | n.s. | n.s. | n.s. | n.s. | n.s. | n.s. | n.s. |
| | SMR per day | n.s. | n.s. | n.s. | n.s. | n.s. | n.s. | n.s. | n.s. | n.s. | n.s. | n.s. | n.s. |
| Day76 | SMR | n.s. | n.s. | n.s. | n.s. | n.s. | n.s. | n.s. | n.s. | n.s. | n.s. | n.s. | n.s. |
| | SMR per day | n.s. | n.s. | n.s. | n.s. | n.s. | n.s. | n.s. | n.s. | n.s. | n.s. | n.s. | n.s. |

SMR: specified SOC mineralization rate Wet: the wet site, Dry: the dry site, LS: limestone soil, AS: acid igneous rock soil, A: A horizon, B: B horizon, LM: large macroaggregates (>2 mm), SM: small macroaggregates (0.25-2mm).

[Figure]

**Fig. R2 Principal component analysis.** DA: α, ω-dioic acid, ω-HA: ω-hydroxyl alkanoic acid, Alkyl: *n*-alkanes and *n*-alkenes, Ps: polysaccharides, N: nitrogen containing compounds, FA<20 Sat: saturated fatty acids with <20 carbon atoms, FA Uns: unsaturated fatty acids, FA20-32: saturated fatty acids with 20-32 carbon atoms, Wet: the wet site, Dry: the dry site, LS: limestone soil, AS: acid igneous rock soil, A: A horizon, B: B horizon. Arrows in solid line mean relative abundance change after incubation of intact aggregates; arrows in dotted line mean relative abundance change after incubation of crushed aggregates.

---

## Referee Comment (RC2) · Anonymous Referee #2 · 30 Sep 2019

The paper discusses the role of lithology and climate on the stabilization of organic matter. I like the choice of the sites on a clear precipitation transect. The approach is also straightforward, but I am not sure why the authors in contrast to the prevailing literature on the topic did not use wet sieving. After all, dry sieving does not result in water stable aggregates that occlude ( to a certain extent) the organic matter. This choice for dry sieving needs to be justified and its implications discussed. Furthermore, details on the dry sieving method are lacking (line 159): agitation intensity and duration. Were the samples air-dried or field moist? The discussion section is speculative as many characteristics are mentioned in the discussion but neither the analytical methods nor the results are presented.

[Figure]

line 103 Could you please explain the land use of the sites in somewhat more detail. As it stands, the land use is grassland, but you also mention cultivation and tree plantations. These activities would belong to cropland or forest land use classes. Line 142 The stoniness is not expressed in % but in fraction. Please also state that you use the gravimetric fraction. See the discussion on the role of coarse fragments for SOC stocks in SOIL by Poeplau et al and Hobley et al (2017 if I am not mistaken). The Bulk density should include the coarse fragments. Was this the case? You mention in line 132 that the gravels were removed. Please revise carefully. Line 144 In general wet sieving is used to determine aggregate stability. Why did you choose dry sieving? Line 147 Please specify that these are gravimetric gravel contents. Lines 307-326 I miss a discussion on the difference between wet and dry sieving. After all, the authors you cite all used wet sieving. It is possible that occlusion does not play an important role, because your aggregates are not water stable and therefore, there is no real occlusion of OM in stable aggregates. This possibility should at least be mentioned in a note of caution (see also general remark). Section 4.3 It is not clear to what extent characteristics have been measured. For instance, lines 328-330 I have not seen any analytical data on Fe and Al hydroxide or Ca bridges. Lines 368-369 How were these fatty acids analysed?

---

## Author Comment (AC2) · 4 Oct 2019

Dear Reviewer:

Thank you for your comments on this manuscript. Your comments really help us to improve the manuscript. We tried to address each point and answer your questions one by one.

**Answers:**

The paper discusses the role of lithology and climate on the stabilization of organic matter. I like the choice of the sites on a clear precipitation transect. The approach is also straightforward, but I am not sure why the authors in contrast to the prevailing literature on the topic did not use wet sieving. After all, dry sieving does not result in water stable aggregates that occlude (to a certain extent) the organic matter. This choice for dry sieving needs to be justified and its implications discussed. Furthermore, details on the dry sieving method are lacking (line 159): agitation intensity and duration. Were the samples air-dried or field moist? The discussion section is speculative as many characteristics are mentioned in the discussion but neither the analytical methods nor the results are presented.

**Answer in general:** Thank you for the comments. In general, we used dry sieving instead of wet sieving because (1) density fraction (or wet-sieving plus ultrasound) can cause serve OM dispersion for acidic soils (ASs), and (2) using wet sieving to determine aggregate stability seems not to contribute to the topic of the manuscript. For detailed information, please check the answer to the question of **Lines 307-326** (isolating occluded OM) and that of **Line 144** (aggregate stability).

For other points, air-dried soils rather than moist soils were applied for the dry sieving. The sieving agitation intensity and duration were just the same as mentioned in the previous description of dry-sieving methods (Line 146-147). To make this part more clear, we will add the intensity and the duration in Line 159 (30 Hz for 20 s).

**Line 103** Could you please explain the land use of the sites in somewhat more detail. As it stands, the land use is grassland, but you also mention cultivation and tree plantations. These activities would belong to cropland or forest land use classes.

**Answer:** The sampling rules were mentioned in **Line 121-123.** All sampling sites had the land use types of grassland, grassland with shrubs or fallow cropland. Tree plantation was avoided because tree litter can induce strong soil acidification. We will re-organize the sentences related to land use to make them easy to understand.

The reason why we include three land use types is that a previous study in this area found that the spatial distribution of SOC stocks is not controlled by land use (Yang et al., 2018). The limited effects of land use on SOC stocks may be attributed to the special land use strategy in which a cycle of cultivation, land set-aside and grazing were repeated every 3-5 years. This suggests the SOC sequestration might be in a dynamical balance. Thus, it is reasonable to sample from these land use types.

**Line 142** The stoniness is not expressed in % but in fraction. Please also state that you use the gravimetric fraction. See the discussion on the role of coarse fragments for SOC stocks in SOIL by Poeplau et al and Hobley et al (2017 if I am not mistaken). The Bulk density should include the coarse fragments. Was this the case? You mention in line 132 that the gravels were removed. Please revise carefully.

**Answer:** Thank you for the very good questions and the relevant references. We apologize for the unclear statement of bulk density. We agree that the bulk density should include the coarse fragments and we actually have included all coarse fragments for the bulk density determination. The weights of coarse fragments were used to revise the bulk density for the SOC stock calculation because the coarse fragments were considered free of organic carbon.

We will make changes in **line 132** to emphasize that bulk densities were measured with coarse fragments involved and SOC contents were measured without coarse fragments involved.

We will also make changes for the formula as follows:

$$SOC\ stock = \sum_{i=1}^{i=k} BD_i \times C_i \times (1 - S_i) \times D_i$$

In which, $BD_i$ = bulk density (g cm$^{-3}$) of the layer i (including gravels), $C_i$ = SOC content (%) of the layer i (excluding gravels), $S_i$ = stoniness of layer i, $D_i$ = thickness (cm) of layer i.

**Line 144** In general wet sieving is used to determine aggregate stability. Why did you choose dry sieving?

**Answer:** We agree that wet sieving is more suitable to determine aggregate stability than dry sieving. We also have the dataset of aggregates stability, macroaggreagte stability determined using wet sieving and microaggregate stability determined using sonication and sedimentation, respectively (details in Fig. R1). However, the objectives of this paper were to have insights into aggregate-size distribution and the stability of SOC distributed in different-sized aggregates. For these objectives, we considered that wet sieving is less suitable than dry sieving for two reasons.

The first reason is that we need to apply incubation to estimate SOC stability in different-sized aggregates. Compared to wet sieving, dry sieving is less destructive and keep the aggregates more similar to the original statues. The second reason is that aggregate stability determined by wet sieving seem not significantly contribute to the paper's topic. Literature showed that aggregate stability is not very useful to estimate SOC stability or OM occluded in aggregates (e.g. Heckman et al., 2014). This is also indicated by our data that neither macroaggregate stability nor microaggregate stability significantly predicted SOC mineralization (Fig. R1). After evaluation, we believe that the aggregate stability determined by wet sieving did not contribute to the topic of this manuscript. Thus, aggregate stability determined by wet sieving was not included.

**Line 147** Please specify that these are gravimetric gravel contents.

**Answer:** We agree to do that.

**Lines 307-326** I miss a discussion on the difference between wet and dry sieving. After all, the authors you cite all used wet sieving. It is possible that occlusion does not play an important role, because your aggregates are not water stable and therefore, there is no real occlusion of OM in stable aggregates. This possibility should at least be mentioned in a note of caution (see also general remark).

**Answer:** Thank you for this question. We will include this point in the paragraph if we have a chance to revise the manuscript.

First, we have to explain why we chose dry sieving. We did not apply density fractionation (or wet-sieving plus sonication) to isolate OM occluded in stable aggregates because the application of ultrasound caused severe dispersion of organic materials into dense solution (NaPT) for the ASs. In addition, separating the dispersed organic materials from the solution was extremely difficult. A similar situation has been reported by Kaiser and Guggenberger (2007). As we could not find a solution for this problem, we chose an alternative method (dry sieving plus incubating intact versus crushed aggregates) applied by Goebel et al. (2009), Juarez et al. (2013) and Wang et al. (2014).

We understand the alternative method estimated OM occluded in both stable aggregates and unstable aggregates. However, our final conclusion supports that there is no real occluded OM because no significant difference in SOC mineralization was found after the incubation between intact and crushed aggregates. Thus, the conclusion was still convincing.

**Section 4.**3 It is not clear to what extent characteristics have been measured. For instance, **lines 328-330** I have not seen any analytical data on Fe and Al hydroxide or Ca bridges.

**Answer:** The data of Fe, Al and Ca has been used as the focus of another paper (*Yang et al. 2019 Revised version submitted to Environmental Earth Science*). Briefly, the OM in the ASs was stabilized by interacting with Fe- and Al-oxides, whereas the OM in the LSs was stabilized by Ca bridges in addition to Fe- and Al-oxides (Table R1). In addition, soil pH values were the key factor controlling OM stabilization mechanisms (Table R1). As the focus of this manuscript was aggregate size distribution and OM stability controlled by aggregates, it could be a better way that we proposed the OM stabilization mechanisms using the previous results (i.e. Fe, Al and Ca) and data from this manuscript (i.e. pH).

**Lines 368-369** How were these fatty acids analysed?

**Answer:** Relative abundances of all mentioned compounds (including fatty acids) were measured using a pyrolysis-GC/MS system. As the data was used for another publication paper (*Yang et al. 2019, under review Geoderma*), we just gave a brief description for the analysis in the subtitle of Fig. S2 as follows:

"Pyrolysis-gas chromatography / mass spectrometry (GC/MS) was applied to estimate the molecular composition of the soil organic matter.  Briefly, milled soil samples were hydrolyzed and methylated

using tetra-methyl-ammonium hydroxide (25 % in water). Afterward, a Curie-point pyrolyzer was used for sample pyrolysis. Helium was used as the carrier gas. Initial temperature was kept at 40 °C for 1 min, followed by heating at the rate of 7 °C min$^{-1}$ until 320 °C sustaining for 15 min. The products of the pyrolysis were analyzed by the GC/MS system. Relative abundance of each compound was calculated as the peak area of the compound divided by the sum of peak areas of all identified compounds."

**Figures and Tables:**

[Figure]

**Fig. R1 SOC mineralization rates predicted by macroaggregate stability and microaggregate stability.** Macroaggregate stability was measured by wet-sieving (20 Hz, 5 min) large macroaggreagtes (>2 mm) and determining the mass of remaining materials >2 mm. Microaggregate stability was determined by comparing the differences in mean weight diameters (MWD, μm) of microaggregates (<0.25 mm) between ultrasonic dispersion (20 W, 10 s) and non-dispersion applied.

**Table R1Correlations between SOC contents and selective extracted fractions, and between pH values and selective extracted fractions.** The table shows the Fe, Al and Ca fractions contribution to SOC stabilization and the controls of soil pH on the Al and Ca fractions.

| | Fe (pyrophosphate extracted) | | Al (pyrophosphate extracted) | | Ca (BaCl$_2$ extracted) | |
|---|---|---|---|---|---|---|
| | Correlation | P | Correlation | P | Correlation | P |
| **Wet-LS (n=11)** | | | | | | |
| **SOC content** | **0.932** | **<0.001** | **0.816** | **0.002** | **0.750** | **0.008** |
| **Wet-AS (n=7)** | | | | | | |
| **SOC content** | **0.687** | **0.088** | **0.736** | **0.059** | 0.185 | 0.691 |
| **All (n=18)** | | | | | | |
| **pH** | 0.063 | 0.805 | **-0.704** | **0.001** | **0.532** | **0.023** |

**References:**

Goebel, M. O., Woche, S. K. and Bachmann, J.: Do soil aggregates really protect encapsulated organic matter against microbial decomposition?, Biologia (Bratisl)., 64(3), 443–448, doi:10.2478/s11756-009-0065-z, 2009.

Heckman, K., Throckmorton, H., Clingensmith, C., González Vila, F. J., Horwath, W. R., Knicker, H. and Rasmussen, C.: Factors affecting the molecular structure and mean residence time of occluded organics in a lithosequence of soils under ponderosa pine, Soil Biol. Biochem., 77, 1–11, doi:10.1016/j.soilbio.2014.05.028, 2014.

Juarez, S., Nunan, N., Duday, A. C., Pouteau, V., Schmidt, S., Hapca, S., Falconer, R., Otten, W. and Chenu, C.: Effects of different soil structures on the decomposition of native andadded organic carbon, Eur. J. Soil Biol., 58, 81–90, doi:10.1016/j.ejsobi.2013.06.005, 2013.

Kaiser, K. and Guggenberger, G.: Distribution of hydrous aluminium and iron over density fractions depends on organic matter load and ultrasonic dispersion, Geoderma, 140(1–2), 140–146, doi:10.1016/j.geoderma.2007.03.018, 2007.

Wang, X., Cammeraat, E. L. H., Cerli, C. and Kalbitz, K.: Soil aggregation and the stabilization of organic carbon as affected by erosion and deposition, Soil Biol. Biochem., 72, 55–65, doi:10.1016/j.soilbio.2014.01.018, 2014.

Yang, S., Cammeraat, E., Jansen, B., den Hann, M., van Loon, E. and Recharte, J.: Soil organic carbon stocks controlled by lithology and soil depth in a Peruvian alpine grassland of the Andes, Catena, 171(June), 11–21, doi:10.1016/j.catena.2018.06.038, 2018.

---

## Author Response (AR2)

Dear Topical Editor:

Thank you for your feedback on our manuscript. We gave answers to the comments from the reviewers and made corrections in the manuscript. The underlined numbers after '**Line**' are the line numbers in the revised manuscript. All revised sentences and paragraphs are marked in red in the manuscript.

Yours sincerely

Songyu Yang and coauthors
* * *
**Respond to 1st reviewer's comments:**

Thank you for the thorough revision and the balanced response. Just two minor comments, the troubled with POM separation might have been solvable using a higher density (e.g. 1.8 g*mL-1), and even when slowly rewetting chances are high to get a drying-rewetting CO2 flush which can be assumed to be substantially different between the soils.

**Answer:** Thank you for your kind reply and additional comments. We appreciate getting a possible solution for the problem of POM separation using density fractionation. We will definitely try this in future research. In terms of the $CO_2$ flush caused by the slow rewetting, we will pay attention to this in the future. A possible solution is that adding a control experiment to investigate to what extent this $CO_2$ flush can bias the results of the incubation between different soils.

**Respond to 2nd reviewers' comments:**

The authors have clearly responded to my comments on the previous version. I just noted that the suggestion of reviewer 1 (to which I agree) to reduce the number of abbreviations by writing the name of the soils in full (rather than using As and Ls) was only followed in the abstract. Why do not you use the full name of the soil types throughout the paper?

**Answer:** Thank you for your feedback. We changed all abbreviations (i.e. LSs and ASs) to their full names (i.e. limestone soils and acid igneous rock soils) throughout the manuscript. We also modified some sentences to improve their readability in **Line** (**207-209**, **258-260**, **263-264** and **284-285 and 315**)).

**Changes in manuscript**

(All line numbers corresponds to the revised manuscript)

**Changes in manuscript:**

1. We changed all abbreviations (i.e. LSs and ASs) to their full names (i.e. limestone soils and acid igneous rock soils) throughout the manuscript.

Wet-LSs were changed to limestone soils in/of the wet site; Dry-LSs were changed to limestone soils in/of the dry site; Wet-ASs were modified to acid igneous rock soils in/of the wet site; and Dry-ASs were modified to acid igneous rock soils in/of the dry site.

2. Because of the replacement of the abbreviations, we also modified some sentences to improve their readability. Changes were in **Line (207-209, 258-260, 263-264, 284-285 and 315**)

**Others:**

We made a modification for the title to Table 1 from "…with significant higher SOC mineralization…" to "…with significant higher SOC mineralization rates…".

[revised manuscript text omitted]

---

## Author Response (AR3)

**Respond to reviewers' comments**

**1. Major revision, 31 October 2019**

Dear Reviewers:

Thank you for your comments on this manuscript. We tried to address each point and answer your questions one by one. Figures and Tables can be found at the end of the answers.

The numbers after '**Line**' are the line numbers in the manuscript **before revision**, whereas the underlined number in the bracket are the line numbers in the manuscript **after revision.** All revised sentences and paragraphs are marked in red in the manuscript.

**Respond to 1st reviewer's comments:**

**Line 20 (20) and general** - To improve readability I would use no abbreviations for the soils, there is enough space to use e.g. "limestone soils" instead of LS.

**Answer:** Yes. We replaced LSs and ASs by limestone soils and acid igneous rock soils in the abstract.

**Line 30 (31)** How much occluded OM was present in these soils? Thus is aggregation at all relevant for OM storage in these soils in contrast to mineral association?

**Answer**: We have no idea of the amount of occluded OM because the OM could not be separated using density fractionation for the acid igneous rock soils (ASs). The main reason is that the application of ultrasound caused severe dispersion of organic materials into dense solution (NaPT, 1.6 g cm$^{-3}$) for the ASs. We tried many ways (including long-time centrifuge) to separate the organic materials but always failed. The dispersion might be attributed to that Na$^+$ in the solution interacted with Al-OM complexes in the ASs and produced a stable suspension. A similar situation has been reported by Kaiser and Guggenberger (2007) but no solution was given.

Thus, we had to choose an alternative method (dry sieving plus incubating intact versus crushed aggregates) applied by Goebel et al. (2009), Juarez et al. (2013) and Wang et al. (2014). Aggregate destruction in the method homogenized soils and reduce the differences in substrate availability in SOC mineralization (Hartley et al., 2007). This way is indirect but still widely used to estimate OM protected by aggregates. As we found no difference in SOC mineralization between intact and crushed aggregates, we proposed that occlusion in aggregates was not the major mechanism for OM stabilization in our soils.

We added a discussion in **Line (323-332)** to explain this.

**Line 40 (41)** The Andes stretch over 7000 km, please be more specific on the location of your work.

**Answer:** To clarify the exact location, we modified the text to emphasized that the large SOC stocks were especially found in the Ecuadorian and Peruvian Andes in **Line (41-44).**

**Line 44-47 (46-48)** Again, the Andes stretch over vast distances, its clear that there are drastic climatic differences. But even at one location you get changes with exposition and elevation. Please be more specific!

**Answer:** We added information in this sentence to specify the study area, Andes in Northern Peru.

**Line 50-51(52-53)** This is redundant in itself, OM and OC is stabilized and of course is tightly linked.

**Answer:** We corrected the sentence to: "Specifically, SOC persistence and stabilization are controlled by…".

**Line 54-56 (57)** Again, this is highly dependent where you are in the Andes. In the Southern Andes you0ll have soils that are completely dominated by particulate OM rather than mineral-associated OM.

**Answer:** We specified this with: **"Peruvian and Ecuadorian Andes".**

**Line 93 (97-100)** describe shortly Puna and Jalca

**Answer:** We modified the sentence to include information of wet Puna and Jalca: "The study areas belong to the Neotropical alpine grassland of the Andes, corresponding to the grassland ecosystem commonly referred as wet Puna or Jalca that is present between the tree line (3500 m asl) and the ice-covered region, having precipitation above 500 mm". More information is referred to the cited paper and the following contents in the M&M.

**Line 103 (110)** Are there records about a longer consistent land use at the sampling sites? Or do you have indices that show a longer sustained land use type?

**Answer:** Please see our answer to the next question (question of **Line 122**).

**Line 122 (131-134)** So there was a mixture of different land uses between the three site replicates? Was this detectable in the soil profiles or SOM properties?

**Answer:** Yes, samples were collected from grassland and abandoned cropland. A previous study showed that SOC stocks were not clearly affected by land use type (e.g. grassland vs. cultivation) (Yang et al. 2018). This is because the local farmers applied a special rotation to change land use in the order of cultivation, abandoned cultivation, cultivate grassland and grassland in a period of several years. They repeated this cycle and it may keep the SOC stock high and in a dynamic balance. We added this information in the sentence in **Line (131-132)**.

**Line 147 (158-160)** How was the gravel content calculated, thus how did the authors differentiate between large aggregates (>2mm) and stones of this size range?

**Answer:** The aggregate-size fractionation was conducted by dry-sieving. For the fraction >5 mm and the fraction 2-5 mm, gravels (stones) were separated by sieving (2 mm) a sub-sample of the fraction after breaking aggregates. The gravel contents were calculated by gravimetric gravel contents, using the gravel weight divided by the sum of the fraction weight plus the gravel weight. In addition, gravels were also excluded in the calculations of mean weight diameters (MWD) and the incubation experiment.

We modified the sentence to "For all fractions larger than 2 mm, gravels were separated by sieving (2 mm) a subsample of the fraction after breaking aggregates. The gravel content (gravimetric) of each fraction was calculated using the gravel weight divided by the sum of the fraction weight plus the gravel weight." in **Line (158-160).**

**Line 157-158 (173-174)** Why you analyse in the one approach the fraction <63 µm, but don't use it in the incubation? Please describe here.

**Answer:** The finer fraction (<0.25 mm) were by far less abundant, especially for the limestone soils. The fraction was therefore not incorporated in the analysis. We added this in the revised manuscript.

**Line 162 (175-176)** By this approach you are not only crushing aggregates, but also rock fragments. How did you account for the different content of pure mineral constituents in relation to aggregates?

**Answer:** We replaced the word "grinding" with "crushing" here (**Line 176**), as this is actually what we did. If rock fragments are defined as size >2mm, this approach will not break rock fragments because rock fragments were removed before the aggregate crushing. We cannot account for the differences between mineral particles and aggregates. Nevertheless, our purpose was only crushing aggregates. Grinding using a porcelain mortar is unlikely to destruct mineral particles.

**Line 164 (178)** It was shown before that aggregate/soil disruption can lead to a rather fast spike in CO2 evolution within a few days. Did you in any way account for this CO2 loss between the different treatments during the first days of incubation before entering a sort of basal respiration?

**Answer:** Thank you for this question. We applied a 10-day pre-incubation, necessary because microbes in the air-dried soils need to be activated. We used slow wetting because we wanted to avoid aggregate destruction caused by fast-wetting.

For the spike in $CO_2$ at the beginning, data from Fig. R1 indicated that the fast pulse in $CO_2$ was not missed. This is because soils were too dry for microbes to start degradation during the early period of the pre-incubation. Furthermore, the comparisons in Table R1 showed no clear differences in early $CO_2$ production between intact and crushed aggregates. Thus, we think that we did not miss the fast $CO_2$ spike during the first few days, whereas our results do not support that aggregate/soil disruption caused a fast spike in $CO_2$ evolution at the beginning.

**Line 208-215 (223-232)** This is a nice exemplary paragraph to show how hard a text is to read for an outsider - let me summarize: "...LS is larger than AS, has more LM but minor Mi; AS has larger SM and Mi; LS were not different but wet-AS was slightly different from dry-AS..." I would really appreciate if you find a way to use even short words that are more descriptive and don0t ruin the flow of reading.

**Answer:** Thank you for this point. We improved the sentences to improve readability. Please read our modified paragraph in **Line (223-232)** in the result section 3.2**.**

**Line 226-227 (243-244)** Please also give mineralization rates normalized to the amount of OC in the individual samples. This will give a better mechanistic insight on the fate of OM with respect to aggregation. This might also level off possible differences in stone content etc.

**Answer:** We already have used SOC mineralization normalized for OC contents. We modified the relevant sentence in the method section **Line (186-187):** "Specific SOC mineralization rates (g CO2-C $g^{-1}$ C), which were normalized for OC contents, were used as an indicator of the C stability of the soil fractions".

**Line 233-234 (250-251)** How is this relation if you normalize OC mineralization rates with sample amount OC?

**Answer:** See our previous answer to the question **Line 226-227** (SOC mineralization already normalized).

**Line 239 (256)** Also if normalized on the amount of OC in LM vs. SM?

**Answer:** See our previous answer to the question **Line 226-227** (SOC mineralization already normalized).

**Line 253-256 (270-272)** You are using two very contrasting parent materials which foster completely different soil biological communities and soil chemistry and thus of course yield different soil structure - so far its textbook knowledge. Such statement might be more interesting if comparing Granodiorite and a Granite or Basalt etc. However, this comment is just about leaving out such "general textbook statements" and focus on the core of the story.

**Answer:** The statements concerning the effects of lithology on soil aggregate size distribution will be largely removed from the discussion part and moved, in a modified form, to the introduction in **Line (72-78)**.

**Line 256-260 (270-272)** This could possibly find its way into the Introduction as you could put this as a rational to take these two contrasting materials. In the discussion it appears again as a redundant textbook message.

**Answer:** We agree to move this part to the introduction in a modified form in **Line (72-78)**.

**Line 262 (moved to 75-77)** So basically the lack of fine material causes the lack of a more advanced aggregation.

**Answer:** This is correct.

**Line 271(280)** You are comparing a silicate rock and a carbonate rock - I would be more than surprised if precipitation would not have a less pronounced effect.

**Answer:** Based on this comment and the next comment concerning stoniness, we considered to shorten the discussion on precipitation in **Line (280-286)** and added an additional discussion on stoniness in **Line (287-298)**.

**Line 275- 280 (287-298)** There are in parts differences in aggregation and SOC stocks between wet and dry sites. Why are you neglecting those and talking them down as minor or biased by stoniness? If stoniness is the driving property, than how can you compare aggregate mineralization etc. at all?

**Answer:** Thank you for the very good questions. We agree that stoniness is a driving factor rather than a minor factor for aggregate-size distribution (Fig. 3). SOC stocks were also slightly higher (not significant) in the wet-ASs compared to the dry-ASs (Fig. 2). However, physicochemical properties of each aggregate

fraction were not clearly affected by stoniness, and thus aggregate mineralization etc. can be compared. We incorporated discussions related to stoniness in **Line (292-298)**. Please check our detailed explanation as follows:

1. SOC stocks are affected by stoniness because we can see from the following equation that SOC stocks get lower when stoniness gets higher.

$$SOC\ stock = \sum_{i=1}^{i=k} BD_i \times C_i \times (1 - S_i) \times D_i$$

*In which, $BD_i$ = bulk density (g cm$^{-3}$) of the layer i (including gravels), $C_i$ = SOC content (%) of the layer i (excluding gravels), $S_i$ = stoniness of layer i, $D_i$= thickness (cm) of layer i.*

2. Mean weight diameters (MWDs, i.e. aggregate-size distribution) gets lower when stoniness gets higher. As stones (gravels) are only distributed in fraction of >5 mm and that of 2-5 mm, the stoniness only has effects on $W_{>5} \times X_{>5}$ and $W_{2-5} \times X_{2-5}$. When the stoniness changed from 0% to 100%, the contribution of $W_{>5} \times X_{>5}$ and $W_{2-5} \times X_{2-5}$ declined and MWDs also decreased. Because the SOC distribution in aggregate fractions coincided with the aggregate-size distribution (Fig. 3), the SOC distribution affected by stoniness was also similar to aggregate-size distribution.

*MWD = $W_{>5} \times X_{>5}$+ $W_{2-5} \times X_{2-5}$+ $W_{0.25-2} \times X_{0.25-2}$+ $W_{0.063-0.25} \times X_{0.063-0.25}$+ $W_{<0.063} \times X_{<0.063}$*

*In which, $Xi$ = averaged diameter (mm) of the fraction i, $Wi$ = weight percent (excluding gravels) of the fraction i.*

3. The SOC mineralization in different aggregate fractions can be compared because the properties of each fraction were not clearly affected by stoniness. Stones (gravels) were excluded in the aggregate-size fractions and in the incubation experiments. The general idea came from the calculation of SOC stocks, in which gravels are considered SOC-free blocks or voids. Detailed explanation can be found in the 2[nd] reviewer's comments **Line 142** and relevant publications (Hobley et al., 2018; Poeplau et al., 2017). As analyses including SOC mineralization were conducted without gravels, we only need to ensure comparable gravel-free soil fractions. Our results showed that properties of aggregate fractions were not clearly affected by stoniness, as indicated by (1) no clear differences in vertical distribution of aggregate-related soil properties between wet-ASs and dry-ASs (Fig. 4) and (2) no clear differences in properties of aggregate fractions between wet-ASs and dry-ASs (Fig. S1). Thus, SOC mineralization rates using aggregate fractions were not biased by gravel contents.

Finally, we believe that it is necessary to discuss this in the discussion part in Line (**287-298**).

**Line 282 (289-290)** Given the high amount of stones and a some other constraints, the significant effects are worth taking them serious. Presumably as a result of altered soil biology and/or plant diversity / litter/root input.

**Answer:** In addition to the answer to the previous question of **Line 275-280**, we could elaborate on the effects of differences in stoniness: root distribution will be different in stones, differences in soil moisture redistribution affecting soil microbial activity and organic matter turnover. We added more explicitly discussions in **Line (289-290)**.

**Line 294-295 (309-310)** Which is a function of primary production and decomposition. Please give in the M&M more details on vegetation at the respective sites.

**Answer:** Information on vegetation were given in the M&M section in **Line (110-114)**: "The vegetation in the wet site is a typical disturbed wet Puna (or Jalca) vegetation with dominant grass species: *Calamagrostis sp.,* but also *Festuca and Agrostis sp. as well as Rumex sp. on fallow land.* Similarly, the vegetation in the dry site is also a typical disturbed wet Puna (or Jalca) vegetation with *Calamogrostis sp., Stipa and Festuca sp. and Rumex sp. on fallow land"*.

**Line 299-304 (314-319)** What soil horizons comprise the low SOC values with high CO2 evolution? Are those the low C/N ratio subsoils? If so, you are mixing two opposite factors, aggregation and soil material origin. Please give specific OC mineralization normalized per amount OC. And the very low C/N ratios under 5, would mean you have pure amino acid material in the sample. Could here values around the detection limit for N play a role?

**Answer:** The mineralization rates were already normalized (Questions of **Line 226-227, Line 233-234** and Line 239). In general, Dry-LS-A had the highest $CO_2$ evolution, whereas Dry-LS-B had the lowest SOC contents. For the C/N ratios, the values were 9.34±0.52 for the Dry-LS-A and 6.86±1.14 for the Dry-LS-B (Fig. 2). Thus, soil horizons with the highest $CO_2$ productions were not subsoils or the horizons with the lowest C/N ratios. In addition, we don't think the N contents reached the detection limit of the Elementar Analyzer because the detection limit of the Analyzer was 0.01% but the lowest N content was 0.16%.

**Line 310-314 (334-338)** How much OM is stored within the aggregates? Do you have estimates of amounts of e.g. occluded POM?

**Answer:** See our answer to the question of **Line 30** and also the question of **Line 307-326** from Reviewer 2.

**Line 315 (339)** The cited work showed a clear effect of aggregate disruption within the first days of incubation. You lack this information due to the late start after 10 days. So the low differences between crushed and intact might be due to fact that you missed the CO2 spike. Furthermore, how did you adjust

comparable soil porosity/O2 diffusion and thus water contents between finely crushed/ground soil material and naturally aggregated soil?

**Answer:** We totally agree that there is a fast spike in $CO_2$ at the beginning of the incubation. However, we had to re-wet the air-dried soils to initiate the decomposition. We choose to slowly re-wet soil materials for 10 days because fast-wetting can significantly break soil aggregates. We just would like to avoid unnecessary destruction of aggregates. At the first few days of the incubation, soil materials are very dry and the SOC mineralization did not start. Thus, the fast spike in $CO_2$ did not appear in this period.

Although we applied the pre-incubation, we believe that we did not miss the massive CO2 production at the beginning. This is because of the much higher CO2 production rates in the first few days of the measurement (Fig. R1). In many studies, the pre-incubations were 14 days. Luckily, we anticipated the fast spike in CO2 at the beginning and we try to shorten the pre-incubation time. If we pre-incubated soils for 14 days as many studies did, we would be more likely missing the CO2 spike that was found in Day 1 and Day 2 (Fig. R1).

For the adjustments of soil porosity and $O_2$ diffusion, we did not make them similar for crushed vs. naturally aggregated soils. OM stabilization through occluded in aggregates can be explained by physical inaccessibility to the decomposer. The inaccessibility is closely related to the microstructure of aggregates (e.g. soil porosity and $O_2$ diffusion). The objectives of crushing aggregates were to destruct soil structure (i.e. soil porosity, $O_2$ diffusion, etc.) that promote OM stabilization. If we made soil porosity and $O_2$ diffusion similar between intact and crushed aggregates, we were a bit like trying to eliminate what we want to compare.

**Line 343(343)** Do you have data on exchangeable ions?

**Answer:** Data on exchangeable ions is the focus of another paper (Yang, in revision, Envir. Earth Scie.). The paper investigated the effects of exchangeable ions, Fe and Al on SOM stabilization. Briefly, the OM in the ASs was stabilized by interacting with Fe- and Al-oxides, whereas the OM in the LSs was stabilized by Ca bridges in addition to Fe- and Al-oxides (Table R2). In addition, soil pH values were the key factor controlling OM stabilization mechanisms (Table R2). As the focus of this manuscript was aggregate size distribution and OM stability controlled by aggregates, it could be a better way that we proposed the OM stabilization mechanisms using the previous results (i.e. Fe, Al and Ca) and data from this manuscript (i.e. pH).

**Line 358 (383)** How is the vegetation at the sites, how is primary production, above and belowground OM input? The biggest control on SOC stocks besides soil properties are plant traits at comparable parent materials. So as stated above, please give information on vegetation data in M&M.

**Answer:** Information on vegetation were given in the M&M section in **Line (110-114).** See the answer to the question of **Line 294-295.** Based on the information, the vegetation is similar between the wet and the dry sites.

**Line 368-370 (392-396)** Or these compounds are just more stable at dry conditions. On top of that, plants produce e.g. more suberin in the roots as protection against drought. And without a baseline of the initial plant material above and belowground this data just tells you there are differences in these acids due to precipitation.

**Answer:** Addressing your first point whether these compounds are more stable in the dry site, our unpublished data showed that they are more vulnerable in the dry-LSs. This is evidenced by the Dry-LSs having a clearer trend in the depletion in $\alpha$, $\omega$-dioic acids and $\omega$-hydroxyl alkanoic acids (maybe also long-chain fatty acids) than the Wet-LSs (Fig. R2). If these compounds are larger and meanwhile more vulnerable in the Dry-LSs compared to the Wet-LSs, the most probable explanation is that the dry-LSs have higher belowground OM input. As it is very difficult to estimate OM input in the puna grassland, we can only assess these potential differences using the data of SOM composition.

We added Fig. R2 in the supplement as Fig. S3 and modified the sentences in **Line (392-396).**

**Line 376-377 (402-403)** There is the same amount of work showing plant species and traits having these effects on SOC storage and stability. Thus to prove the solely precipitation effect you would have to work with comparable plant species and traits.

**Answer:** The vegetation between the two sites is slightly different, but consists of grasses of the same functional types and genera but with different (sub-)species (see M&M in **Line (110-114) ).**Thus, we proposed that their impact on the soil is comparable. With regard to the primary production, we have no data and literature on this is also very scarce but we expect that NPP is also affected by the availability of moisture.

For more open discussion, it is impressive that vegetation has been reported having limited influence on SOC storage and chemical composition in Andean alpine grasslands. Tonneijck et al. (2010) and Zimmermann et al. (2009) showed that SOC stocks were not significantly different between forest and grassland in Ecuadorian and Peruvian Andes. Furthermore, molecular composition was also not clearly different between forest and grassland soils (Nierop et al., 2007). We are not sure whether this is unique for the Andes. This is an interesting topic for future research and also the reason that we focused on soil mineralogy and aggregates.

**Line 381 (407)** So how high is the OM input?

**Answer:** We do not know the exact OM input because it is very difficult to estimate OM input in the Andean Puna/Jalca grassland. In addition, literature on OM production or NPP is very limited. We only found that one publication for Peru on slightly drier sites indicates a NPP of about 5 Mg C / ha yr for grazed grassland and around 15 Mg C /ha yr (Oliveras et al., 2014), which might give an indication of the NPP at our sites. Thus, we have to emphasize that our statement is based on estimation rather than quantitative measurement. We added this point in the previous paragraph in **Line (399-400).**

Nevertheless, we believe that it is a good opportunity that future studies focus on finding a practical method to estimate OM input in the Andean grasslands.

**Line 385 (411)** You compare limestone with granodiorite, as mentioned above this of course outcompetes any effect of precipitation at same altitude and latitude.

**Answer:** We rephrased it by adding the sentence: "We did not find an important effect of precipitation on aggregation, which was probably overshadowed by the effect of lithology." in **Line (411-413).**

**Line 385-387 (413-415)** For this you would have to show that there is no occluded light fraction/POM, and you didn0t miss the fast pulses (>10 days) in CO2 after soil structure disruption found by others.

**Answer:** As density fractionation was not applicable for the acid rock soils, incubation with aggregate intact vs. crushed was used as an alternative method to estimate occluded OM. For the fast pulses, data from Fig. R1 indicated that the fast pulse in $CO_2$ was not missed, whereas the comparisons in Table R1 indicated that SOC production in the first days was not significantly higher for crushed aggregates than intact aggregates. Thus, we can propose that SOM is unlikely stabilized by occluded in aggregates. For details, please check the answers to the questions of **Line** 30, **Line** 164 and **Line** 315.

**Figures and Tables:**

**Table R1 Comparisons in SOC mineralization rates (per day) between intact aggregates and crushed aggregates**

| | | Wet-LS-A | | Wet-LS-B | | Wet-AS-A | | Dry-LS-A | | Dry-LS-B | | Dry-AS-A | |
|---|---|---|---|---|---|---|---|---|---|---|---|---|---|
| | | LM | SM | LM | SM | LM | SM | LM | SM | LM | SM | LM | SM |
| Day1 | SMR | n.s. | n.s. | n.s. | n.s. | n.s. | n.s. | n.s. | n.s. | n.s. | n.s. | n.s. | n.s. |
| | SMR per day | n.s. | n.s. | n.s. | n.s. | n.s. | n.s. | n.s. | n.s. | n.s. | n.s. | n.s. | n.s. |
| Day2 | SMR | n.s. | n.s. | n.s. | n.s. | n.s. | n.s. | n.s. | n.s. | n.s. | n.s. | n.s. | n.s. |
| | SMR per day | n.s. | n.s. | n.s. | n.s. | n.s. | n.s. | n.s. | n.s. | n.s. | n.s. | n.s. | n.s. |
| Day6 | SMR | n.s. | n.s. | n.s. | n.s. | n.s. | n.s. | n.s. | n.s. | n.s. | n.s. | n.s. | n.s. |
| | SMR per day | n.s. | n.s. | n.s. | n.s. | n.s. | n.s. | n.s. | n.s. | n.s. | n.s. | n.s. | n.s. |
| Day9 | SMR | n.s. | n.s. | n.s. | n.s. | n.s. | n.s. | n.s. | n.s. | n.s. | n.s. | n.s. | n.s. |
| | SMR per day | n.s. | n.s. | n.s. | n.s. | n.s. | n.s. | n.s. | n.s. | n.s. | n.s. | n.s. | n.s. |
| Day13 | SMR | n.s. | n.s. | n.s. | n.s. | n.s. | n.s. | n.s. | n.s. | n.s. | n.s. | n.s. | n.s. |
| | SMR per day | n.s. | n.s. | n.s. | n.s. | n.s. | n.s. | n.s. | n.s. | n.s. | n.s. | n.s. | **In>Cr\*\*** |
| Day20 | SMR | n.s. | n.s. | n.s. | n.s. | n.s. | n.s. | n.s. | n.s. | n.s. | n.s. | n.s. | n.s. |

| | | | | | | | | | | | | | |
|---|---|---|---|---|---|---|---|---|---|---|---|---|---|
| | SMR per day | n.s. | n.s. | | n.s. | n.s. | | n.s. | n.s. | | n.s. | n.s. | | n.s. | n.s. | | n.s. | n.s. |
| Day28 | SMR | n.s. | n.s. | | n.s. | n.s. | n.s. | n.s. | n.s. | n.s. | n.s. | n.s. | n.s. | n.s. |
| | SMR per day | n.s. | n.s. | n.s. | n.s. | n.s. | n.s. | n.s. | n.s. | n.s. | n.s. | n.s. | n.s. |
| Day48 | SMR | n.s. | n.s. | n.s. | n.s. | n.s. | n.s. | n.s. | n.s. | n.s. | n.s. | n.s. | n.s. |
| | SMR per day | n.s. | n.s. | n.s. | n.s. | n.s. | n.s. | n.s. | n.s. | n.s. | n.s. | n.s. | n.s. |
| Day76 | SMR | n.s. | n.s. | n.s. | n.s. | n.s. | n.s. | n.s. | n.s. | n.s. | n.s. | n.s. | n.s. |
| | SMR per day | n.s. | n.s. | n.s. | n.s. | n.s. | n.s. | n.s. | n.s. | n.s. | n.s. | n.s. | n.s. |

SMR: specified SOC mineralization rate Wet: the wet site, Dry: the dry site, LS: limestone soil, AS: acid igneous rock soil, A: A horizon, B: B horizon, LM: large macroaggregates (>2 mm), SM: small macroaggregates (0.25-2mm).

[Figure]

**Fig. R1 Specific SOC mineralization rate per day (g C mineralized g$^{-1}$ SOC day$^{-1}$).** Wet: the wet site, Dry: the dry site, LS: limestone soil, AS: acid igneous rock soil, A: A horizon, B: B horizon, LM: large macroaggregates (>2 mm), SM: small macroaggregates (0.25-2mm).

**Table R2 Correlations between SOC contents and selective extracted fractions, and between pH values and selective extracted fractions.** The table shows the Fe, Al and Ca fractions contribution to SOC stabilization and the controls of soil pH on the Al and Ca fractions.

| | Fe (pyrophosphate extracted) | | Al (pyrophosphate extracted) | | Ca (BaCl$_2$ extracted) | |
|---|---|---|---|---|---|---|
| | Correlation | P | Correlation | P | Correlation | P |
| **Wet-LS (n=11)** | | | | | | |

| SOC content Wet-AS (n=7) | 0.932 | <0.001 | 0.816 | 0.002 | 0.750 | 0.008 |
|---|---|---|---|---|---|---|
| SOC content All (n=18) | 0.687 | 0.088 | 0.736 | 0.059 | 0.185 | 0.691 |
| pH | 0.063 | 0.805 | -0.704 | 0.001 | 0.532 | 0.023 |

[Figure]

**Fig. R2 (Fig. S3) Principal component analysis.** DA: α, ω-dioic acid, ω-HA: ω-hydroxyl alkanoic acid, Alkyl: *n*-alkanes and *n*-alkenes, Ps: polysaccharides, N: nitrogen containing compounds, FA<20 Sat: saturated fatty acids with <20 carbon atoms, FA Uns: unsaturated fatty acids, FA20-32: saturated fatty acids with 20-32 carbon atoms, Wet: the wet site, Dry: the dry site, LS: limestone soil, AS: acid igneous rock soil, A: A horizon, B: B horizon. Arrows in solid line mean relative abundance change after incubation of intact aggregates; arrows in dotted line mean relative abundance change after incubation of crushed aggregates.

**Respond to 2nd reviewers' comments:**

The paper discusses the role of lithology and climate on the stabilization of organic matter. I like the choice of the sites on a clear precipitation transect. The approach is also straightforward, but I am not sure why the authors in contrast to the prevailing literature on the topic did not use wet sieving. After all, dry sieving does not result in water stable aggregates that occlude (to a certain extent) the organic matter. This choice for dry sieving needs to be justified and its implications discussed. Furthermore, details on the dry sieving method are lacking (line 159): agitation intensity and duration. Were the samples air-dried or field moist? The discussion section is speculative as many characteristics are mentioned in the discussion but neither the analytical methods nor the results are presented.

**Answer in general:** Thank you for the comments. In general, we used dry sieving instead of wet sieving because that (1) the method using wet-sieving to isolate occluded OM is not applicable for the acidic soils (ASs), and (2) aggregate stability determined by wet sieving does not explain SOC stocks or stability and is not helpful for to answer our research questions. For detailed information, please check the answer to the question of **Lines 307-326** (isolating occluded OM) and the question of **Line 144** (aggregate stability).

For other points, air-dried soils rather than moist soils were applied for the dry sieving. The sieving agitation intensity and duration were just the same as mentioned in the previous description of dry-sieving methods in **Line (155-161)**. To make this part clearer, we added the intensity and the duration in Line **(172)** "(30 Hz for 20 s)". For other soil characteristics, they were the focus of another publication. We addressed this in the question of **Line 328-330.**

**Line 103 (109)** Could you please explain the land use of the sites in somewhat more detail. As it stands, the land use is grassland, but you also mention cultivation and tree plantations. These activities would belong to cropland or forest land use classes.

**Answer:** The sampling rules were mentioned in **Line (131-134).** All sampling sites had the land use types of grassland, grassland with shrubs or fallow cropland. Tree plantation was avoided because tree litter can induce strong soil acidification.

The reason why we include three land use types is that a previous study in this area found that the spatial distribution of SOC stocks is not controlled by land use (Yang et al., 2018). The limited effects of land use on SOC stocks may be attributed to the special land use strategy in which a cycle of cultivation, land set-aside and grazing were repeated every 3-5 years. This suggests the SOC sequestration might be in a dynamical balance. Thus, it is reasonable to sample from these land use types.

We added information on vegetation in **Line (111-114)** and explained why we sampled on these land use types in **Line (131-132)**.

**Line 142 (153)** The stoniness is not expressed in % but in fraction. Please also state that you use the gravimetric fraction. See the discussion on the role of coarse fragments for SOC stocks in SOIL by

Poeplau et al and Hobley et al (2017 if I am not mistaken). The Bulk density should include the coarse fragments. Was this the case? You mention in line 132 that the gravels were removed. Please revise carefully.

**Answer:** Thank you for the very good questions and the relevant references. We apologize for the unclear statement of bulk density. We read the recommended publications and agree that the bulk density should include the coarse fragments. We actually have included all coarse fragments for the bulk density determination. The weights of coarse fragments were used to revise the bulk density for the SOC stock calculation because the coarse fragments were considered free of organic carbon.

We made changes in **Line (143-146)** to emphasize that bulk densities were measured with coarse fragments involved and SOC contents were measured without coarse fragments involved. The changes were: "Soil samples collected every 10 cm were freeze-dried to determine bulk densities and SOC stocks. Soil bulk densities were measured by weighing samples after freeze-drying. Afterward, gravels (>2 mm) were removed from the samples. The rest of the samples were used to determine OC contents and to calculate SOC stocks."

We also made changes for the formula in **Line (152-154)** as follows:

$$SOC\ stock = \sum_{i=1}^{i=k} BD_i \times C_i \times (1 - S_i) \times D_i$$

In which, $BD_i$ = bulk density (g cm$^{-3}$) of the layer i (including gravels), $C_i$ = SOC content (%) of the layer i (excluding gravels), $S_i$ = stoniness (gravimetric) of layer i, $D_i$ = thickness (cm) of layer i.

**Line 144 (155)** In general wet sieving is used to determine aggregate stability. Why did you choose dry sieving?

**Answer:** We agree that wet sieving is more suitable to determine aggregate stability than dry sieving. We also have the dataset of aggregates stability, macroaggreagte stability determined using wet sieving and microaggregate stability determined using sonication and sedimentation, respectively (details in Fig. R3). However, the objectives of this paper were to have insights into aggregate-size distribution and the stability of SOC distributed in different-sized aggregates. For these objectives, we considered that wet sieving is less suitable than dry sieving for two reasons.

The first reason is that we need to apply incubation to estimate SOC stability in different-sized aggregates. Compared to wet sieving, dry sieving is less destructive and keep the aggregates more similar to the original statues. The second reason is that aggregate stability determined by wet sieving seem not significantly contribute to the paper's topic. Literature showed that aggregate stability is not very useful to estimate SOC stability or OM occluded in aggregates (e.g. Heckman et al., 2014). This is also indicated by our data that neither macroaggregate stability nor microaggregate stability significantly predicted SOC mineralization (Fig. R3). After evaluation, we believe that the aggregate stability determined by wet

sieving did not contribute to the topic of this manuscript. Thus, aggregate stability determined by wet sieving was not included.

We added a discussion to explain why we have chosen dry-sieving instead of wet-sieving in **Line (323-332)**.

**Line 147 (158-160)** Please specify that these are gravimetric gravel contents.

**Answer:** Corrected. We modified the sentence to "Gravels (>2mm) were removed for all fractions larger than 2mm and the gravel content (gravimetric) of each fraction was calculated using the gravel weight divided by the sum of the fraction weight plus the gravel weight." in **Line (158-160)**.

**Lines 307-326 (322-332)** I miss a discussion on the difference between wet and dry sieving. After all, the authors you cite all used wet sieving. It is possible that occlusion does not play an important role, because your aggregates are not water stable and therefore, there is no real occlusion of OM in stable aggregates. This possibility should at least be mentioned in a note of caution (see also general remark).

**Answer:** Thank you for this question. We added a discussion in **Line (323-332)** to explain the application of dry-sieving plus incubation instead of wet-sieving plus sonication. Please also see the detailed explanation as follows.

First, we have to explain why we chose dry sieving. In general, wet-sieving is used to get water-stable aggregates in which OM is occluded and stabilized. In order to isolate and quantify the occluded OM in water-stable aggregates (what we need), density fractionation plus sonication is generally applied. We had no problem for the wet-sieving but had problems for the sonication. The application of ultrasound caused severe dispersion of organic materials into dense solution (NaPT) for the acidic soils (ASs). The dispersed organic materials were extremely difficult to be isolated from the solution. A similar situation has been reported by Kaiser and Guggenberger (2007), but we could not find a solution. It is not enough to get insights into aggregate-protected OM if we only quantified water-stable aggregates. Thus we have to use an alternative method to estimate occluded OM.

We chose an alternative method (dry sieving plus incubating intact versus crushed aggregates) applied by Goebel et al. (2009) and Wang et al. (2014). In this method, OM occluded in water-stable aggregates and unstable aggregates was addressed. Our conclusion supports that aggregate occlusion did not clear promote SOC stabilization. This can be explained by either no occluded OM or occluded OM being not stabilized.

**Section 4.3** It is not clear to what extent characteristics have been measured. For instance, **lines 328-330 (351-354)** I have not seen any analytical data on Fe and Al hydroxide or Ca bridges.

**Answer:** The data of Fe, Al and Ca has been used as the focus of another paper (*Yang et al. 2019 Revised version submitted to Environmental Earth Science*). Briefly, the OM in the ASs was stabilized by interacting with Fe- and Al-oxides, whereas the OM in the LSs was stabilized by Ca bridges in addition to Fe- and Al-oxides (Table R2). In addition, soil pH values were the key factor controlling OM stabilization mechanisms (Table R2). As the focus of this manuscript was aggregate size distribution and OM stability controlled by aggregates, it could be a better way that we proposed the OM stabilization mechanisms using the previous results (i.e. Fe, Al and Ca) and data from this manuscript (i.e. pH).

**Lines 368-369 (392-394)** How were these fatty acids analysed?

**Answer:** Relative abundances of all mentioned compounds (including fatty acids) were measured using a pyrolysis-GC/MS system. As the data was used for another publication paper (*Yang et al. 2019, under review in Geoderma*), we just gave a brief description to the analysis in the subtitle of Fig. S2 as follows:

"Pyrolysis-gas chromatography / mass spectrometry (GC/MS) was applied to estimate the molecular composition of the soil organic matter.  Briefly, milled soil samples were hydrolyzed and methylated using tetra-methyl-ammonium hydroxide (25 % in water). Afterward, a Curie-point pyrolyzer was used for sample pyrolysis. Helium was used as the carrier gas. Initial temperature was kept at 40 °C for 1 min, followed by heating at the rate of 7 °C min$^{-1}$ until 320 °C sustaining for 15 min. The products of the pyrolysis were analyzed by the GC/MS system. Relative abundance of each compound was calculated as the peak area of the compound divided by the sum of peak areas of all identified compounds."

**Figures and Tables:**

[Figure]

**Fig. R3 SOC mineralization rates predicted by macroaggregate stability and microaggregate stability.** Macroaggregate stability was measured by wet-sieving (20 Hz, 5 min) large macroaggreagtes (>2 mm) and determining the mass of remaining materials >2 mm. Microaggregate stability was determined by comparing the differences in mean weight diameters (MWD, μm) of microaggregates (<0.25 mm) between ultrasonic dispersion (20 W, 10 s) and non-dispersion applied.

**Table R2Correlations between SOC contents and selective extracted fractions, and between pH values and selective extracted fractions.** The table shows the Fe, Al and Ca fractions contribution to SOC stabilization and the controls of soil pH on the Al and Ca fractions.

|  | Fe (pyrophosphate extracted) | | Al (pyrophosphate extracted) | | Ca (BaCl$_2$ extracted) | |
|---|---|---|---|---|---|---|
|  | Correlation | P | Correlation | P | Correlation | P |
| **Wet-LS (n=11)** | | | | | | |
| **SOC content** | 0.932 | <0.001 | 0.816 | 0.002 | 0.750 | 0.008 |
| **Wet-AS (n=7)** | | | | | | |
| **SOC content** | 0.687 | 0.088 | 0.736 | 0.059 | 0.185 | 0.691 |
| **All (n=18)** | | | | | | |
| **pH** | 0.063 | 0.805 | -0.704 | 0.001 | 0.532 | 0.023 |

**References:**

Goebel, M. O., Woche, S. K. and Bachmann, J.: Do soil aggregates really protect encapsulated organic matter against microbial decomposition?, Biologia (Bratisl)., 64(3), 443–448, doi:10.2478/s11756-009-0065-z, 2009.

Hartley, I. P., Heinemeyer, A. and Ineson, P.: Effects of three years of soil warming and shading on the rate of soil

respiration: Substrate availability and not thermal acclimation mediates observed response, Glob. Chang. Biol., 13(8), 1761–1770, doi:10.1111/j.1365-2486.2007.01373.x, 2007.

Heckman, K., Throckmorton, H., Clingensmith, C., González Vila, F. J., Horwath, W. R., Knicker, H. and Rasmussen, C.: Factors affecting the molecular structure and mean residence time of occluded organics in a lithosequence of soils under ponderosa pine, Soil Biol. Biochem., 77, 1–11, doi:10.1016/j.soilbio.2014.05.028, 2014.

Hobley, E. U., Murphy, B. and Simmons, A.: Comment on "soil organic stocks are systematically overestimated by misuse of the parameters bulk density and rock fragment content" by Poeplau et al. (2017), Soil, 4(2), 169–171, doi:10.5194/soil-4-169-2018, 2018.

Juarez, S., Nunan, N., Duday, A. C., Pouteau, V., Schmidt, S., Hapca, S., Falconer, R., Otten, W. and Chenu, C.: Effects of different soil structures on the decomposition of native andadded organic carbon, Eur. J. Soil Biol., 58, 81–90, doi:10.1016/j.ejsobi.2013.06.005, 2013.

Kaiser, K. and Guggenberger, G.: Distribution of hydrous aluminium and iron over density fractions depends on organic matter load and ultrasonic dispersion, Geoderma, 140(1–2), 140–146, doi:10.1016/j.geoderma.2007.03.018, 2007.

Nierop, K. G. J., Tonneijck, F. H., Jansen, B. and Verstraten, J. M.: Organic Matter in Volcanic Ash Soils under Forest and Páramo along an Ecuadorian Altitudinal Transect, Soil Sci. Soc. Am. J., 71(4), 1119, doi:10.2136/sssaj2006.0322, 2007.

Oliveras, I., Girardin, C., Doughty, C. E., Cahuana, N., Arenas, C. E., Oliver, V., Huaraca Huasco, W. and Malhi, Y.: Andean grasslands are as productive as tropical cloud forests, Environ. Res. Lett., doi:10.1088/1748-9326/9/11/115011, 2014.

Poeplau, C., Vos, C. and Don, A.: Soil organic carbon stocks are systematically overestimated by misuse of the parameters bulk density and rock fragment content, Soil, 3(1), 61–66, doi:10.5194/soil-3-61-2017, 2017.

Tonneijck, F. H., Jansen, B., Nierop, K. G. J., Verstraten, J. M., Sevink, J. and De Lange, L.: Towards understanding of carbon stocks and stabilization in volcanic ash soils in natural Andean ecosystems of northern Ecuador, Eur. J. Soil Sci., 61(3), 392–405, doi:10.1111/j.1365-2389.2010.01241.x, 2010.

Wang, X., Cammeraat, E. L. H., Cerli, C. and Kalbitz, K.: Soil aggregation and the stabilization of organic carbon as affected by erosion and deposition, Soil Biol. Biochem., 72, 55–65, doi:10.1016/j.soilbio.2014.01.018, 2014.

Yang, S., Cammeraat, E., Jansen, B., den Hann, M., van Loon, E. and Recharte, J.: Soil organic carbon stocks controlled by lithology and soil depth in a Peruvian alpine grassland of the Andes, Catena, 171(June), 11–21, doi:10.1016/j.catena.2018.06.038, 2018.

Zimmermann, M., Meir, P., Silman, M. R., Fedders, A., Gibbon, A., Malhi, Y., Urrego, D. H., Bush, M. B., Feeley, K. J., Garcia, K. C., Dargie, G. C., Farfan, W. R., Goetz, B. P., Johnson, W. T., Kline, K. M., Modi, A. T., Rurau, N. M. Q., Staudt, B. T. and Zamora, F.: No Differences in Soil Carbon Stocks Across the Tree Line in the Peruvian Andes, Ecosystems, 13(1), 62–74, doi:10.1007/s10021-009-9300-2, 2009.

**2. Minor revision, 02 December 2019**

Dear Topical Editor:

Thank you for your feedback on our manuscript. We gave answers to the comments from the reviewers and made corrections in the manuscript. The underlined numbers after '**Line**' are the line numbers in the revised manuscript. All revised sentences and paragraphs are marked in red in the manuscript.

Yours sincerely

Songyu Yang and coauthors
* * *
**Respond to 1ˢᵗ reviewer's comments:**

Thank you for the thorough revision and the balanced response. Just two minor comments, the troubled with POM separation might have been solvable using a higher density (e.g. 1.8 g*mL-1), and even when slowly rewetting chances are high to get a drying-rewetting CO2 flush which can be assumed to be substantially different between the soils.

**Answer:** Thank you for your kind reply and additional comments. We appreciate getting a possible solution for the problem of POM separation using density fractionation. We will definitely try this in future research. In terms of the $CO_2$ flush caused by the slow rewetting, we will pay attention to this in the future. A possible solution is that adding a control experiment to investigate to what extent this $CO_2$ flush can bias the results of the incubation between different soils.

**Respond to 2ⁿᵈ reviewers' comments:**

The authors have clearly responded to my comments on the previous version. I just noted that the suggestion of reviewer 1 (to which I agree) to reduce the number of abbreviations by writing the name of the soils in full (rather than using As and Ls) was only followed in the abstract. Why do not you use the full name of the soil types throughout the paper?

**Answer:** Thank you for your feedback. We changed all abbreviations (i.e. LSs and ASs) to their full names (i.e. limestone soils and acid igneous rock soils) throughout the manuscript. We also modified some sentences to improve their readability in **Line** (**207-209**, **258-260**, **263-264** and **284-285 and 315**)).